# Fast Extra Gradient Methods for Smooth Structured Nonconvex-Nonconcave Minimax Problems

**Sucheol Lee**
Department of Mathematical Sciences
KAIST
Daejeon, Republic of Korea
csfh1379@kaist.ac.kr

**Donghwan Kim**
Department of Mathematical Sciences
KAIST
Daejeon, Republic of Korea
donghwankim@kaist.ac.kr

## Abstract

Modern minimax problems, such as generative adversarial network and adversarial training, are often under a nonconvex-nonconcave setting, and developing an efficient method for such setting is of interest. Recently, two variants of the extragradient (EG) method are studied in that direction. First, a two-time-scale variant of the EG, named EG+, was proposed under a smooth structured nonconvex-nonconcave setting, with a slow $\mathcal{O}(1/k)$ rate on the squared gradient norm, where $k$ denotes the number of iterations. Second, another variant of EG with an anchoring technique, named extra anchored gradient (EAG), was studied under a smooth convex-concave setting, yielding a fast $\mathcal{O}(1/k^2)$ rate on the squared gradient norm. Built upon EG+ and EAG, this paper proposes a two-time-scale EG with anchoring, named fast extragradient (FEG), that has a fast $\mathcal{O}(1/k^2)$ rate on the squared gradient norm for smooth structured nonconvex-nonconcave problems; the corresponding saddle-gradient operator satisfies the negative comonotonicity condition. This paper further develops its backtracking line-search version, named FEG-A, for the case where the problem parameters are not available. The stochastic analysis of FEG is also provided.

## 1 Introduction

Recently, nonconvex-nonconcave minimax problems have received an increased attention in the optimization community and the machine learning community due to their applications to generative adversarial network [10] and adversarial training [27]. In this paper, we consider a smooth structured nonconvex-nonconcave minimax problem:

$$\min_{\boldsymbol{x} \in \mathbb{R}^{d_x}} \max_{\boldsymbol{y} \in \mathbb{R}^{d_y}} f(\boldsymbol{x}, \boldsymbol{y}), \tag{1}$$

where $f : \mathbb{R}^{d_x} \times \mathbb{R}^{d_y} \to \mathbb{R}$ is smooth and is possibly nonconvex in $\boldsymbol{x}$ for fixed $\boldsymbol{y}$, and possibly nonconcave in $\boldsymbol{y}$ for fixed $\boldsymbol{x}$; the saddle-gradient operator $\boldsymbol{F} := (\nabla_x f, -\nabla_y f)$ satisfies the negative comonotonicity [1]. We construct an efficient (first-order) method, using a saddle gradient operator $\boldsymbol{F}$ for finding a first-order stationary point of the problem (1).

So far little is known under the nonconvex-nonconcave setting, compared to the convex-concave setting. Recent works [4, 7, 22, 24, 26, 42, 44] studied extragradient-type methods [19, 39] for minimax problems under various structured nonconvex-nonconcave settings. In other words, they consider various non-monotone conditions on $\boldsymbol{F}$, such as the Minty variational inequality (MVI) condition [4], the weak MVI condition [7], and the negative comonotonicity [1].[1] Among them, this

35th Conference on Neural Information Processing Systems (NeurIPS 2021).

---

[1]Relations between the conditions on $\boldsymbol{F}$ considered in this paper is summarized in Figure 1.

paper focuses on the negative comonotonicity condition for a Lipschitz continuous $\boldsymbol{F}$. To the best of our knowledge, the following two-time-scale variant of the extragradient method, named EG+:

$$
\begin{aligned}
\boldsymbol{z}_{k+1/2} &= \boldsymbol{z}_k - \frac{\alpha_k}{\beta} \boldsymbol{F} \boldsymbol{z}_k, \\
\boldsymbol{z}_{k+1} &= \boldsymbol{z}_k - \alpha_k \boldsymbol{F} \boldsymbol{z}_{k+1/2},
\end{aligned}
\tag{EG+}
$$

is the only known (explicit)[2] method, using $\boldsymbol{F}$, that converges under the considered setting[3] [7], where $\boldsymbol{z}_k := (\boldsymbol{x}_k, \boldsymbol{y}_k)$. The EG+, however, has a slow $\mathcal{O}(1/k)$ rate on the squared gradient norm. Note that a similar two-time-scale approach has been found to stabilize the stochastic extragradient method with unbounded noise variance [14].

Meanwhile, under the smooth convex-concave setting, recent works [6, 17, 21, 40, 43] suggest that Halpern-type [12] (or anchoring) methods, performing a convex combination of an initial point $\boldsymbol{z}_0$ and the last updated point $\boldsymbol{z}_k$ at each iteration, has a fast $\mathcal{O}(1/k^2)$ rate in terms of the squared gradient norm. In particular, [43] developed the following anchoring variant of the extragradient method, named extra anchored gradient (EAG):

$$
\begin{aligned}
\boldsymbol{z}_{k+1/2} &= \boldsymbol{z}_k + \beta_k(\boldsymbol{z}_0 - \boldsymbol{z}_k) - \alpha_k \boldsymbol{F} \boldsymbol{z}_k, \\
\boldsymbol{z}_{k+1} &= \boldsymbol{z}_k + \beta_k(\boldsymbol{z}_0 - \boldsymbol{z}_k) - \alpha_k \boldsymbol{F} \boldsymbol{z}_{k+1/2}.
\end{aligned}
\tag{EAG}
$$

This is the first (explicit) method with a fast $\mathcal{O}(1/k^2)$ rate on the squared gradient norm, when $\boldsymbol{F}$ satisfies both the Lipschitz continuity and the monotonicity. [43] also showed that such $\mathcal{O}(1/k^2)$ rate is optimal for first-order methods using a Lipschitz continuous and monotone $\boldsymbol{F}$.

Built upon both EG+ and EAG, this paper studies the following class of two-time-scale anchored extragradient methods, named fast extragradient (FEG):

$$
\begin{aligned}
\boldsymbol{z}_{k+1/2} &= \boldsymbol{z}_k + \beta_k(\boldsymbol{z}_0 - \boldsymbol{z}_k) - (1 - \beta_k)(\alpha_k + 2\rho_k)\boldsymbol{F} \boldsymbol{z}_k, \\
\boldsymbol{z}_{k+1} &= \boldsymbol{z}_k + \beta_k(\boldsymbol{z}_0 - \boldsymbol{z}_k) - \alpha_k \boldsymbol{F} \boldsymbol{z}_{k+1/2} - (1 - \beta_k)2\rho_k \boldsymbol{F} \boldsymbol{z}_k.
\end{aligned}
\tag{Class FEG}
$$

Note that (Class FEG) reuses the $\boldsymbol{F} \boldsymbol{z}_k$ term in the $\boldsymbol{z}_{k+1}$ update, unlike the standard extragradient-type methods, which we found essential for handling the negative comonotonicity condition. We leave further understanding the use of $\boldsymbol{F} \boldsymbol{z}_k$ and the formulation of (Class FEG) as future work. The proposed FEG method (with appropriately chosen step coefficients $\alpha_k$, $\beta_k$ and $\rho_k$ discussed later) has an $\mathcal{O}(1/k^2)$ rate on the squared gradient norm, under the Lipschitz continuity and the negative comonotonicity conditions on $\boldsymbol{F}$. To the best of our knowledge, this is the first accelerated method under the nonconvex-nonconcave setting. The FEG also has value under the smooth convex-concave setting. First, when $\boldsymbol{F}$ is Lipschitz continuous and monotone, the rate bound of FEG is about 27/4 times smaller than that of EAG. Also note that the rate bound of FEG is only about four times larger than the $\mathcal{O}(1/k^2)$ lower complexity bound of first-order methods under such setting [43], further closing the gap between the lower and upper complexity bounds. Second, when $\boldsymbol{F}$ is cocoercive, FEG has a rate faster than that of a version of Halpern iteration [12] in [6].

We also develop an adaptive variant of FEG, named FEG-A, which updates its parameters, $\alpha_k$ and $\rho_k$ in (Class FEG), adaptively using a backtracking line-search [2, 25, 31]. FEG requires the knowledge of the two problem parameters for the Lipschitz continuity and the comonotonicity of $\boldsymbol{F}$. However, those global parameters can be conservative, and in practice, they are even usually unknown. For such cases, the FEG-A adaptively and locally estimates the problem parameters, while preserving the fast rate $\mathcal{O}(1/k^2)$ on the squared gradient norm for smooth structured nonconvex-nonconcave minimax problems.

Lastly, we study a stochastic version of FEG, named S-FEG, which uses an unbiased stochastic estimate of $\boldsymbol{F} \boldsymbol{z}$, i.e., $\tilde{\boldsymbol{F}} \boldsymbol{z} = \boldsymbol{F} \boldsymbol{z} + \xi$, instead of $\boldsymbol{F} \boldsymbol{z}$ in FEG, where $\xi$ denotes a stochastic noise. For a Lipschitz continuous and monotone $\boldsymbol{F}$, we provide a convergence analysis in terms of the expected squared gradient norm. In specific, we show that the S-FEG is stable with a rate $\mathcal{O}(1/k^2) + \mathcal{O}(\epsilon)$, when the noise variance decreases in the order of $\mathcal{O}(\epsilon/k)$, while being unstable otherwise due to

---

[2] A proximal point method converges under the negative comonotonicity [1, 18], but such implicit method is not preferable over explicit methods in practice due to its implicit nature.

[3] The EG+ was originally shown to work under the weak MVI condition of $\boldsymbol{F}$, which is weaker than the negative comonotonicity.

error accumulation. This is similar to the convergence behavior of a stochastic version of Nesterov's fast gradient method [35, 36], observed in [5], for smooth convex minimization.

Our main contributions are summarized as follows.

- We propose the FEG method that has an accelerated convergence rate $\mathcal{O}(1/k^2)$ on the squared gradient norm for smooth structured nonconvex-nonconcave minimax problems.

- We present that the FEG method has a rate faster than those of the EAG and the Halpern iteration for smooth convex-concave problems.

- We construct a backtracking line-search version of FEG, named FEG-A, for the case where the Lipschitz constant and comonotonicity parameters of $\boldsymbol{F}$ are unavailable.

- We analyze a stochastic version of FEG, named S-FEG, for smooth convex-concave problems.

## 2 Related work

### 2.1 Methods for convex-concave minimax problems

The extragradient method [19] is one of the widely used methods for solving smooth convex-concave minimax problems (see, *e.g.*, [4, 7, 22, 24, 26, 42, 44] for its extensions and applications). In terms of the *duality gap*, $\max_{\boldsymbol{y}' \in \mathcal{Y}} f(\boldsymbol{x}, \boldsymbol{y}') - \min_{\boldsymbol{x}' \in \mathcal{X}} f(\boldsymbol{x}', \boldsymbol{y})$, where $\mathcal{X}$ and $\mathcal{Y}$ are compact[4] domains, the ergodic iterate of the extragradient-type methods [32, 37] have an $\mathcal{O}(1/k)$ rate. Such $\mathcal{O}(1/k)$ rate on the duality gap is order-optimal for the first-order methods [34, 38], leaving no room for improvement. On the other hand, the last iterate of the extragradient method has a slower $\mathcal{O}(1/\sqrt{k})$ rate on the duality gap, under an additional assumption that $\boldsymbol{F}$ has a Lipschitz derivative [9]. In terms of the *squared gradient norm*, $\|\boldsymbol{F}\boldsymbol{z}\|^2$, the best iterate of the extragradient-type methods [19, 39] have an $\mathcal{O}(1/k)$ rate [40, 41, 43]. The last iterate of the extragradient method also has a rate $\mathcal{O}(1/k)$, when $\boldsymbol{F}$ is further assumed to have a Lipschitz derivative [9]. Unlike the duality gap, the $\mathcal{O}(1/k)$ rate on the squared gradient norm is not optimal [43]. From now on throughout this paper, we mainly study and compare the convergence rates on the squared gradient norm, which still has room for improvement in convex-concave problems, and has meaning for nonconvex-nonconcave minimax problems, unlike the duality gap.

Recently, [6, 17, 21, 40, 43] found that Halpern-type [12] (or anchoring) methods yield a fast $\mathcal{O}(1/k^2)$ rate in terms of the squared gradient norm for minimax problems. [17, 21] showed that the (implicit) Halpern iteration [12] with appropriately chosen step coefficients has an $\mathcal{O}(1/k^2)$ rate on the squared norm of a monotone $\boldsymbol{F}$. Then, for a cocoercive $\boldsymbol{F}$, an (explicit) version of the Halpern iteration was studied in [6, 17] that has the same fast rate. In addition, [6] constructed a double-loop version of the Halpern iteration for a Lipschitz continuous and monotone $\boldsymbol{F}$, which has a rate $\tilde{\mathcal{O}}(1/k^2)$ on the squared gradient norm, slower than the rate $\mathcal{O}(1/k^2)$. While this is promising compared to the $\mathcal{O}(1/k)$ rate of the extragradient methods on the squared gradient norm [40, 41, 43], the computational complexity due to its double-loop nature and a relatively slow rate remained a problem. Very recently, [43] proposed the extra anchored gradient (EAG) method, which is the first (explicit) method with a fast $\mathcal{O}(1/k^2)$ rate for smooth convex-concave minimax problems, *i.e.*, for Lipschitz continuous and monotone operators. In addition, [43] proved that the EAG is order-optimal by showing that the lower complexity bound of first-order methods is $\Omega(1/k^2)$.

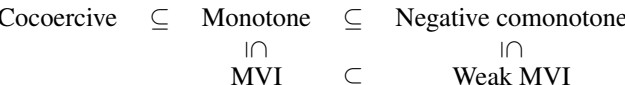

Figure 1: Relations between the conditions on $\boldsymbol{F}$.

---

[4]The convergence analysis on the duality gap of the extragradient type methods are generalized under the unbounded domain assumption in [28, 29, 30].

## 2.2 Methods for nonconvex-nonconcave minimax problems

Some recent literature considered relaxing the monotonicity condition of the saddle gradient operator to tackle modern nonconvex-nonconcave minimax problems. For example, the Minty variational inequality (MVI) condition, *i.e.*, there exists $\boldsymbol{z}_* \in \boldsymbol{Z}_*(\boldsymbol{F})$ satisfying $\langle \boldsymbol{F}\boldsymbol{z}, \boldsymbol{z} - \boldsymbol{z}_* \rangle \geq 0$ for all $\boldsymbol{z} \in \mathbb{R}^d$ where $\boldsymbol{Z}_*(\boldsymbol{F}) := \{\boldsymbol{z}_* \in \mathbb{R}^d : \boldsymbol{F}\boldsymbol{z}_* = \boldsymbol{0}\}$, is studied in [4, 23, 22, 24]. This condition is also studied under the name, the coherence, in [26, 42, 44]. Moreover, [7] considered a weaker condition, named the weak MVI condition, *i.e.*, for some $\rho < 0$, there exists $\boldsymbol{z}_* \in \boldsymbol{Z}_*(\boldsymbol{F})$ satisfying $\langle \boldsymbol{F}\boldsymbol{z}, \boldsymbol{z} - \boldsymbol{z}_* \rangle \geq \rho\|\boldsymbol{F}\boldsymbol{z}\|^2$ for all $\boldsymbol{z} \in \mathbb{R}^d$. The weak MVI condition is implied by the negative comonotonicity [1] or, equivalently, the (positive) cohypomonotonicity [3]. The comonotonicity will be further discussed in the upcoming section.

For $L$-Lipschitz continuous $\boldsymbol{F}$, [4, 42] showed that the extragradient-type methods have an $\mathcal{O}(1/k)$ rate on the squared gradient norm under the MVI condition, and [7] developed the (EG+) method under the weak MVI condition (and thus under the negative comonotonicty), which also has an $\mathcal{O}(1/k)$ rate on the squared gradient norm. To the best of our knowledge, there is no known accelerated method for the nonconvex-nonconcave setting; our proposed FEG method is the first method to have a fast $\mathcal{O}(1/k^2)$ rate under the nonconvex-nonconcave setting. The convergence rates of the existing methods and the FEG on the squared gradient norm are summarized in Table 1.

Table 1: Comparison of the convergence rates of the existing extragradient-type methods and the FEG, with respect to the squared gradient norm, for smooth structured minimax problems, under various assumptions on the Lipschitz continuous saddle gradient operator $\boldsymbol{F}$.

| | Method | Convex-concave | | Nonconvex-nonconcave | | |
|---|---|---|---|---|---|---|
| | | Cocoercive | Monotone | Negative comonotone | MVI | Weak MVI |
| Normal | EG [4, 42] | $\mathcal{O}(1/k)$ | $\mathcal{O}(1/k)$ | | $\mathcal{O}(1/k)$ | |
| | EG+ [7] | $\mathcal{O}(1/k)$ | $\mathcal{O}(1/k)$ | $\mathcal{O}(1/k)$ | $\mathcal{O}(1/k)$ | $\mathcal{O}(1/k)$ |
| Accelerated | Halpern [12, 6] | $\mathcal{O}(1/k^2)$ | $\tilde{\mathcal{O}}(1/k^2)$ | | | |
| | EAG [43] | $\mathcal{O}(1/k^2)$ | $\mathcal{O}(1/k^2)$ | | | |
| | FEG (this paper) | $\mathcal{O}(1/k^2)$ | $\mathcal{O}(1/k^2)$ | $\mathcal{O}(1/k^2)$ | | |

## 3 Preliminaries

The followings are the two main assumptions for the saddle gradient operator $\boldsymbol{F}$ of the smooth structured nonconvex-nonconcave problem (1). Under such assumptions, we develop efficient methods that find a first-order stationary point $\boldsymbol{z}_* \in \boldsymbol{Z}_*(\boldsymbol{F})$ where $\boldsymbol{Z}_*(\boldsymbol{F}) := \{\boldsymbol{z}_* \in \mathbb{R}^d : \boldsymbol{F}\boldsymbol{z}_* = \boldsymbol{0}\}$.

**Assumption 1** ($L$-Lipschitz continuity). *For some $L \in (0, \infty)$, $\boldsymbol{F}$ satisfies*

$$\|\boldsymbol{F}\boldsymbol{z} - \boldsymbol{F}\boldsymbol{z}'\| \leq L\|\boldsymbol{z} - \boldsymbol{z}'\|, \quad \forall \boldsymbol{z}, \boldsymbol{z}' \in \mathbb{R}^d.$$

**Assumption 2** ($\rho$-Comonotonicity). *For some $\rho \in \left(-\frac{1}{2L}, \infty\right)$, $\boldsymbol{F}$ satisfies*

$$\langle \boldsymbol{F}\boldsymbol{z} - \boldsymbol{F}\boldsymbol{z}', \boldsymbol{z} - \boldsymbol{z}' \rangle \geq \rho\|\boldsymbol{F}\boldsymbol{z} - \boldsymbol{F}\boldsymbol{z}'\|^2, \quad \forall \boldsymbol{z}, \boldsymbol{z}' \in \mathbb{R}^d.$$

The $\rho$-comonotonicity consists of three cases depending on the choice of $\rho$; the negative comonotonicity when $\rho < 0$, the monotonicity when $\rho = 0$, and the cocoercivity when $\rho > 0$. The negative comonotonicity is weaker than the other two, and is the main focus of this paper. The following is an examplary nonconvex-nonconcave condition that is stronger than the negative comonotonicity [1, 3].

**Example 1.** *Let $f$ be twice continuously differentiable and $\gamma$-weakly-convex-weakly-concave. Further assume that $f$ satisfies*

$$\nabla_{\boldsymbol{xx}}^2 f + \nabla_{\boldsymbol{xy}}^2 f(\eta\boldsymbol{I} - \nabla_{\boldsymbol{yy}}^2 f)^{-1}\nabla_{\boldsymbol{yx}}^2 f \succeq \alpha\boldsymbol{I}, \tag{2}$$

$$-\nabla_{\boldsymbol{yy}}^2 f + \nabla_{\boldsymbol{yx}}^2 f(\eta\boldsymbol{I} + \nabla_{\boldsymbol{xx}}^2 f)^{-1}\nabla_{\boldsymbol{xy}}^2 f \succeq \alpha\boldsymbol{I},$$

*for some $\alpha \geq 0$ and $\eta > \gamma$, named $\alpha \geq 0$-interaction dominant condition in [11]. Then, the saddle gradient of $f$ satisfies the $-\frac{1}{\eta}$-negative comonotonicity. (See Appendix A.1.) For any $\gamma$-weakly-convex-weakly-concave function, the condition (2) holds with $\alpha = -\gamma < 0$. Its extreme case is*

$f(x,y) = -\frac{\gamma}{2}x^2 + \frac{\gamma}{2}y^2$, where there is no interaction between $x$ and $y$. On the other hand, when the the second terms in the left-hand side of (2) are sufficently positive definite, a nonconvex-nonconcave function satisfies the condition (2) with a nonnegative $\alpha$. In specific, the $\alpha \geq 0$-interaction dominant condition is satisfied when the interaction term of Hessian $\nabla^2_{xy}f$ is dominating any negative curvature in Hessians $\nabla^2_{xx}f$ and $-\nabla^2_{yy}f$ [11].

We next present our proposed FEG, and illustrate that the FEG outperforms existing methods such as EG+, EAG, and the Halpern iteration, for each three comonoticity case, respectively.

## 4 Fast extragradient (FEG) method for Lipschitz continuous and comonotone operators

This section considers an instance of (Class FEG) with $\alpha_k = \frac{1}{L}$, $\beta_k = \frac{1}{k+1}$, and $\rho_k = \rho$ for all $k \geq 0$. The resulting method, named FEG, is illustrated in Algorithm 1, which has an $\mathcal{O}(1/k^2)$ fast rate with respect to the squared gradient norm, in Theorem 4.1. The proof of Theorem 4.1 is provided in Section 7.

---

**Algorithm 1** Fast extragradient (FEG) method

---

**Input:** $z_0 \in \mathbb{R}^d$, $L \in (0, \infty)$, $\rho \in \left(-\frac{1}{2L}, \infty\right)$
**for** $k = 0, 1, \ldots$ **do**

$$z_{k+1/2} = z_k + \frac{1}{k+1}(z_0 - z_k) - \left(1 - \frac{1}{k+1}\right)\left(\frac{1}{L} + 2\rho\right)\boldsymbol{F}z_k$$

$$z_{k+1} = z_k + \frac{1}{k+1}(z_0 - z_k) - \frac{1}{L}\boldsymbol{F}z_{k+1/2} - \left(1 - \frac{1}{k+1}\right)2\rho\boldsymbol{F}z_k.$$

**end for**

---

**Theorem 4.1.** *For the $L$-Lipschitz continuous and $\rho$-comonotone operator $\boldsymbol{F}$ with $\rho > -\frac{1}{2L}$ and for any $\boldsymbol{z}_* \in \boldsymbol{Z}_*(\boldsymbol{F})$, the sequence $\{z_k\}_{k \geq 0}$ generated by FEG satisfies, for all $k \geq 1$,*

$$\|\boldsymbol{F}z_k\|^2 \leq \frac{4\|z_0 - z_*\|^2}{\left(\frac{1}{L} + 2\rho\right)^2 k^2}. \tag{3}$$

The following example shows that the bound (3) of the FEG is exact for $\rho = 0$ and $k = 4l + 2$. The bound (3) is not known to be exact in general, and we leave finding the exact bound as future work.

**Example 2.** *Let $f : \mathbb{R} \times \mathbb{R} \to \mathbb{R}$ be $f(x,y) = Lxy$. Its saddle gradient operator and solution are $\boldsymbol{F}(x,y) = (Ly, -Lx)$ and $\boldsymbol{z}_* = (0,0)$, respectively. For the initial point $\boldsymbol{z}_0 = (x_0, y_0) = (1, 0)$, the sequence $\{z_k\}_{k \geq 0}$ generated by FEG satisfies $z_{4l+2} = \left(0, \frac{1}{2l+1}\right)$ for all $l \geq 0$. Hence, $\|\boldsymbol{F}z_{4l+2}\|^2 = \frac{L^2}{(2l+1)^2} = \frac{4L^2\|z_0 - z_*\|^2}{(4l+2)^2}$ for all $l \geq 0$. (See Appendix B.1.)*

We next compare the rate bound (3) with existing analyses for the three cases $-\frac{1}{2L} < \rho < 0$, $\rho = 0$, and $\rho > 0$.

### 4.1 Comparison to EG+ under the negative comonotonicity ($\rho < 0$)

Under the negative comonotonicity with $-\frac{1}{8L} < \rho < 0$, the (EG+) method with $\alpha_k = \frac{1}{2L}$ and $\beta = \frac{1}{2}$ has an $\mathcal{O}(1/k)$ rate on the squared gradient norm. To the best of our knowledge, this is the best known rate, and the FEG has a faster $\mathcal{O}(1/k^2)$ rate with a wider region of convergence $-\frac{1}{2L} < \rho < 0$.

### 4.2 Comparison to EAG under the monotonicity ($\rho = 0$)

For an $L$-Lipschitz continuous and monotone operator $\boldsymbol{F}$, [43] proposed two EAG methods, named EAG-C and EAG-V, with same $\beta_k = \frac{1}{k+2}$ but with different choices of $\alpha_k$. EAG-C sets $\alpha_k$ to be a constant $\frac{1}{8L}$ for all $k \geq 0$ in (EAG), and has a large constant 260 in its convergence rate,

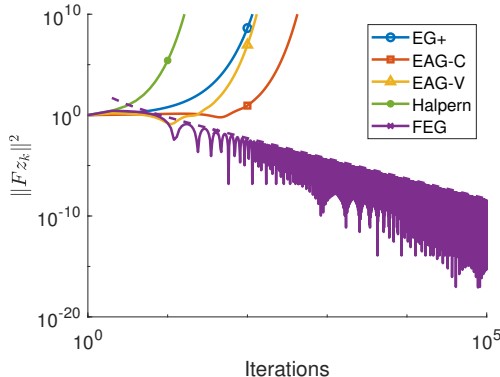

Figure 2: Numerical result with $f(x, y) = -\frac{1}{6}x^2 + \frac{2\sqrt{2}}{3}xy + \frac{1}{6}y^2$. The dashed line represents the theoretical bound (3) of FEG.

$\|\boldsymbol{F}\boldsymbol{z}_k\|^2 \leq \frac{260L^2\|\boldsymbol{z}_0 - \boldsymbol{z}_*\|^2}{(k+1)^2}$ for all $k \geq 0$. On the other hand, while EAG-V requires a complicated recursive update for $\{\alpha_k\}$, $\alpha_{k+1} = \frac{\alpha_k}{1 - \alpha_k^2 L^2}\left(1 - \frac{(k+2)^2}{(k+1)(k+3)}\alpha_k^2 L^2\right)$ for all $k \geq 0$, with $\alpha_0 = \frac{0.618}{L}$, its rate has a smaller constant 27.

The FEG takes a constant $\alpha_k = \frac{1}{L}$, unlike EAG-V, but has an even smaller constant 4 in its convergence rate $\|\boldsymbol{F}\boldsymbol{z}_k\|^2 \leq \frac{4L^2\|\boldsymbol{z}_0 - \boldsymbol{z}_*\|^2}{k^2}$ for $\rho = 0$. Therefore, the FEG with $\rho = 0$ has about $260/4$-times and $27/4$-times faster convergence rate compared to those of EAG-C and EAG-V, respectively. Furthermore, the rate bound of FEG with $\rho = 0$ is only about 4-times larger than the lower complexity bound of first-order methods under the considered setting [43], reducing the gap between the lower and upper complexity bounds from 27 to 4.

### 4.3 Comparison to the Halpern iteration under the cocoercivity ($\rho > 0$)

For a $\rho$-cocoercive operator $\boldsymbol{F}$, an (explicit) version of Halpern iteration [12], studied in [6], has a fast rate, $\|\boldsymbol{F}\boldsymbol{z}_k\|^2 \leq \frac{\|\boldsymbol{z}_0 - \boldsymbol{z}_*\|^2}{\rho^2 k^2}$. Note that while the $\rho$-cocoercivity implies the $\frac{1}{\rho}$-Lipschitz continuity, there is case where the $\rho$-cocoercive (and thus Lipschitz continuous) operator has a Lipschitz constant $L$ smaller than $\frac{1}{\rho}$. Since $L \leq \frac{1}{\rho}$, the FEG has a rate $\|\boldsymbol{F}\boldsymbol{z}_k\|^2 \leq \frac{4\|\boldsymbol{z}_0 - \boldsymbol{z}_*\|^2}{(1/L + 2\rho)^2 k^2} = \frac{4\|\boldsymbol{z}_0 - \boldsymbol{z}_*\|^2}{9\rho^2 k^2}$ that is faster than that of Halpern iteration. However, if we take into account that the FEG requires computing the saddle gradient twice per iteration, unlike Halpern iteration studied in [6], the FEG method has a slower rate in terms of the number of gradient computations. If we narrow down to the case $L < \frac{1}{2\rho}$, the FEG has a faster rate, $\|\boldsymbol{F}\boldsymbol{z}_k\|^2 \leq \frac{4\|\boldsymbol{z}_0 - \boldsymbol{z}_*\|^2}{(1/L + 2\rho)^2 k^2} < \frac{\|\boldsymbol{z}_0 - \boldsymbol{z}_*\|^2}{4\rho^2 k^2}$. For such case, the FEG has a rate faster than that of the Halpern iteration, even in terms of the number of gradient computations.

### 4.4 Toy example

We performed a toy experiment on a simple quadratic function, $f(x, y) = \frac{\rho L^2}{2}x^2 + L\sqrt{1 - \rho^2 L^2}xy - \frac{\rho L^2}{2}y^2$, which has an $L$-Lipschitz continuous and $\rho$-comonotone saddle gradient. For the case $\rho = -\frac{1}{3L}$ and $L = 1$, Figure 2 illustrates that the FEG converges with an accelerated rate whereas EG+, EAG-C, EAG-V, and the (explicit) version of Halpern iteration [6] diverge. This example presents that the existing guarantees on convergence and acceleration of the aforementioned methods under the convex-concave setting do not generalize to the nonconvex-nonconcave setting.

## 5 FEG with backtracking line-search

The FEG requires the knowledge of the two global parameters $L$ and $\rho$ for Lipschitz continuity and comonotonicity, respectively. Those global parameters are often difficult to compute in practice

and can be locally conservative. To handle these two disadvantages, we employ the backtracking line-search technique [2, 25, 31] in FEG. We adaptively decrease the two step size parameters, $\tau$ and $\eta$, to satisfy the both conditions, the local $\frac{1}{\tau}$-Lipschitz continuity and the $\frac{\eta-\tau}{2}$-comonotonicity.[5] A pseudocode of the resulting method, named FEG-A, is illustrated in Algorithm 2. For a detailed description of the FEG-A, see Algorithm 4 in Appendix C.1.

---

**Algorithm 2** Fast extragradient method with adaptive step size (FEG-A)

---

**Input:** $z_0 \in \mathbb{R}^d$, $\tau_{-1} \in (\max\{0, -2\rho\}, \infty)$, $\eta_0 \in (0, \infty)$, $\delta \in (0, 1)$
Find the smallest nonnegative integer $i_0$ such that $\hat{z} = z_0 - \tau_{-1}(1 - \delta)^{i_0} Fz_0$ satisfies $\|F\hat{z} - Fz_0\| \leq \frac{1}{\tau_{-1}(1-\delta)^{i_0}} \|\hat{z} - z_0\|$.
$\tau_0 = \tau_{-1}(1 - \delta)^{i_0}$, $z_1 = z_0 - \tau_0 Fz_0$.
**for** $k = 1, 2, \ldots$ **do**
  $i_k = j_k = 0$.
  Increase each $i_k$ and $j_k$ one by one until

$$\hat{z}_{k+1/2} = z_k + \frac{1}{k+1}(z_0 - z_k) - \left(1 - \frac{1}{k+1}\right)\eta_{k-1}(1 - \delta)^{j_k} Fz_k \qquad \text{and}$$

$$\hat{z}_{k+1} = z_k + \frac{1}{k+1}(z_0 - z_k) - \tau_{k-1}(1 - \delta)^{i_k} Fz_{k+1/2}$$

$$- \left(1 - \frac{1}{k+1}\right)\left(\eta_{k-1}(1 - \delta)^{j_k} - \tau_{k-1}(1 - \delta)^{i_k}\right) Fz_k$$

satisfy both conditions,

$$\|F\hat{z}_{k+1} - F\hat{z}_{k+1/2}\| \leq \frac{1}{\tau_{k-1}(1-\delta)^{i_k}}\|\hat{z}_{k+1} - \hat{z}_{k+1/2}\| \qquad \text{and}$$

$$\langle F\hat{z}_{k+1} - Fz_k, \hat{z}_{k+1} - z_k \rangle \geq \frac{\eta_{k-1}(1 - \delta)^{j_k} - \tau_{k-1}(1 - \delta)^{i_k}}{2}\|F\hat{z}_{k+1} - Fz_k\|^2.$$

  $\tau_k = \tau_{k-1}(1 - \delta)^{i_k}$, $\eta_k = \eta_{k-1}(1 - \delta)^{j_k}$, $z_{k+1} = \hat{z}_{k+1}$.
**end for**

---

The following lemma shows that each of the nonincreasing sequences $\{\tau_k\}_{k \geq 0}$ and $\{\eta_k\}_{k \geq 0}$ of the FEG-A has a positive lower bound, and thus FEG-A is well-defined[6], under the condition $\rho > -\frac{\tau_k}{2}$. This condition for $\rho$ can be weaker than the condition $\rho > -\frac{1}{2L}$ of FEG, since the local Lipschitz parameter $\frac{1}{\tau_k}$ can be smaller than $L$. This is another benefit of using a backtracking line-search in FEG, over the standard FEG.

**Lemma 5.1.** *For the $L$-Lipschitz and $\rho$-comonotone operator $F$ and a given constant $\delta \in (0, 1)$, the step size $\tau_k$ of FEG-A is lower bounded by a positive value $\underline{\tau} := \min\left\{\tau_{-1}, \frac{1-\delta}{L}\right\}$ for all $k \geq 0$, and if $\rho > -\frac{\tau_k}{2}$, the step size $\eta_k$ is lower bounded by a positive value $\min\left\{\eta_0, (1 - \delta)(\tau_k + 2\rho)\right\}$ for all $k \geq 1$.*

The FEG-A method also has the following $\mathcal{O}(1/k^2)$ rate with respect to the squared gradient norm in Theorem 5.1, when $\rho > -\frac{\tau_k}{2}$. The proof is provided in Section 7 and Appendix C.3.

**Theorem 5.1.** *For the $L$-Lipschitz and $\rho$-comonotone operator $F$ and for any $z_* \in Z_*(F)$, the sequence $\{z_k\}_{k \geq 0}$ generated by FEG-A satisfies*

$$\|Fz_k\|^2 \leq \frac{4\|z_0 - z_*\|^2}{((k-1)\eta_k + \tau_k + 2\rho)^2}$$

*for all $k \geq 1$, if $\rho > -\frac{\tau_k}{2}$.*

This rate bound of FEG-A reduces to that of FEG in Theorem 4.1, when we choose $\tau_{-1} = \frac{1}{L}$ and $\eta_0 = \frac{1}{L} + 2\rho$ for FEG-A.

---

[5] In specific, $\tau$ and $\eta$ locally estimate $\frac{1}{L}$ and $\frac{1}{L} + 2\rho$, respectively. One could have directly estimate $\rho$, instead of $\frac{1}{L} + 2\rho$, but this complicates the line-search process to handle both positive and negative values of $\rho$, unlike our choice of $\eta$ in FEG-A.

[6] This requires one to chooses $\tau_{-1}$ strictly greater than the unknown value $-2\rho$ when $\rho < 0$.

## 6 FEG under stochastic setting

When exactly computing $\boldsymbol{Fz}$ is expensive in practice, one usually instead consider its stochastic estimate for computational efficiency (see, *e.g.*, [13, 16, 26, 33, 40, 42, 44]). This section also considers using a stochastic oracle in FEG for smooth convex-concave problems. In specific, this section assumes that we only have access to a noisy saddle gradient oracle, $\tilde{\boldsymbol{F}}\boldsymbol{z}_{k/2} = \boldsymbol{F}\boldsymbol{z}_{k/2} + \xi_{k/2}$, where $\{\xi_{k/2}\}_{k\geq 0}$ are independent random variables satisfying $\mathbb{E}[\xi_{k/2}] = 0$ and $\mathbb{E}[\|\xi_{k/2}\|^2] = \sigma_{k/2}^2$ for all $k \geq 0$. Under this setting, we study a stochastic first-order method, named stochastic fast extragradient (S-FEG) method, illustrated in Algorithm 3.

---

**Algorithm 3** Stochastic fast extragradient (S-FEG) method

---

**Input:** $\boldsymbol{z}_0 \in \mathbb{R}^d$, $L \in (0, \infty)$.
**for** $k = 0, 1, \dots$ **do**

$$\boldsymbol{z}_{k+1/2} = \boldsymbol{z}_k + \frac{1}{k+1}(\boldsymbol{z}_0 - \boldsymbol{z}_k) - \left(1 - \frac{1}{k+1}\right)\frac{1}{L}\tilde{\boldsymbol{F}}\boldsymbol{z}_k$$

$$\boldsymbol{z}_{k+1} = \boldsymbol{z}_k + \frac{1}{k+1}(\boldsymbol{z}_0 - \boldsymbol{z}_k) - \frac{1}{L}\tilde{\boldsymbol{F}}\boldsymbol{z}_{k+1/2}$$

**end for**

---

The following theorem provides an upper bound of the expected squared gradient norm for the S-FEG. (See Appendix D.3 for the proof.)

**Theorem 6.1.** *Let $\tilde{\boldsymbol{F}}\boldsymbol{z}_{k/2} = \boldsymbol{F}\boldsymbol{z}_{k/2} + \xi_{k/2}$, where $\{\xi_{k/2}\}_{k\geq 0}$ are independent random variables satisfying $\mathbb{E}[\xi_{k/2}] = 0$ and $\mathbb{E}[\|\xi_{k/2}\|^2] = \sigma_{k/2}^2$ for all $k \geq 0$. Then, for the $L$-Lipschitz continuous and monotone operator $F$ and for any $\boldsymbol{z}_* \in \boldsymbol{Z}_*(\boldsymbol{F})$, the sequence $\{\boldsymbol{z}_k\}_{k\geq 0}$ generated by S-FEG satisfies*

$$\mathbb{E}[\|\boldsymbol{F}\boldsymbol{z}_k\|^2] \leq \frac{4L^2\|\boldsymbol{z}_0 - \boldsymbol{z}_*\|^2}{k^2} + \frac{6}{k^2}\left[\sigma_0^2 + \sum_{l=1}^{k-1}(l^2\sigma_l^2 + (l+1)^2\sigma_{l+1/2}^2)\right] \tag{4}$$

*for all $k \geq 1$. Furthermore, if $\sigma_0^2 \leq \frac{\epsilon}{6}$, $\sigma_k^2 \leq \frac{\epsilon}{6k}$ and $\sigma_{k+1/2}^2 \leq \frac{\epsilon}{6(k+1)}$ for all $k \geq 1$, then the bound (4) reduces to*

$$\mathbb{E}[\|\boldsymbol{F}\boldsymbol{z}_k\|^2] \leq \frac{4L^2\|\boldsymbol{z}_0 - \boldsymbol{z}_*\|^2}{k^2} + \epsilon$$

*for all $k \geq 1$.*

Here, we needed the noise variance $\sigma_{k/2}^2$ to decrease in the order of $\mathcal{O}(1/k)$ so that the stochastic error of the S-FEG does not accumulate. Otherwise, if $\sigma_{k/2}^2$ is a constant for all $k$, the error accumulates with rate $\mathcal{O}(k)$. In short, the S-FEG will suffer from error accumulation, unless the stochastic error decreases with rate $\mathcal{O}(1/k)$. Such error accumulation behavior also appears in a stochastic version of Nesterov's fast gradient method [35, 36] for smooth convex minimization [5, 8]. Similar to [5], we believe that adjusting the step coefficients of the S-FEG can make the S-FEG become relatively stable even with a constant noise, which we leave as future work.

## 7 Convergence analysis with nonincreasing potential lemma

We analyze FEG and FEG-A by finding a nonincreasing potential function in a form $V_k = a_k\|\boldsymbol{F}\boldsymbol{z}_k\|^2 - b_k\langle\boldsymbol{F}\boldsymbol{z}_k, \boldsymbol{z}_0 - \boldsymbol{z}_k\rangle$ in the lemma below. We provide a similar potential lemma for S-FEG in Appendix D.2. The convergence analyses of EAG and Halpern iteration are also based on such potential function [6, 43].

**Lemma 7.1.** *Let $\{\boldsymbol{z}_k\}_{k\geq 0}$ be the sequence generated by* (Class FEG) *with $\{\alpha_k\}_{k\geq 0}$, $\{\beta_k\}_{k\geq 0}$, $\{L_k\}_{k\geq 0} \subset (0, \infty)$ and $\{\rho_k\}_{k\geq 0} \subset \mathbb{R}$, satisfying $\alpha_0 \in (0, \infty)$, $\alpha_k \in \left(0, \frac{1}{L_k}\right]$, $\beta_0 = 1$, $\{\beta_k\}_{k\geq 1} \subseteq (0, 1)$ for all $k \geq 1$, and*

$$\frac{(1 - \beta_{k+1})}{2\beta_{k+1}}(\alpha_{k+1} + 2\rho_{k+1}) - \rho_{k+1} \leq \frac{1}{2\beta_k}(\alpha_k + 2\rho_k) - \rho_k$$

*for all $k \geq 0$. Assume that the following conditions are satisfied.*

$$\|\boldsymbol{F}\boldsymbol{z}_1 - \boldsymbol{F}\boldsymbol{z}_0\| \leq L_0\|\boldsymbol{z}_1 - \boldsymbol{z}_0\|$$
$$\|\boldsymbol{F}\boldsymbol{z}_{k+1} - \boldsymbol{F}\boldsymbol{z}_{k+1/2}\| \leq L_k\|\boldsymbol{z}_{k+1} - \boldsymbol{z}_{k+1/2}\| \quad \text{for all } k \geq 1,$$
$$\langle \boldsymbol{F}\boldsymbol{z}_{k+1} - \boldsymbol{F}\boldsymbol{z}_k, \boldsymbol{z}_{k+1} - \boldsymbol{z}_k \rangle \geq \rho_k\|\boldsymbol{F}\boldsymbol{z}_{k+1} - \boldsymbol{F}\boldsymbol{z}_k\|^2 \quad \text{for all } k \geq 1.$$

*Then the potential function*

$$V_k = a_k\|\boldsymbol{F}\boldsymbol{z}_k\|^2 - b_k\langle \boldsymbol{F}\boldsymbol{z}_k, \boldsymbol{z}_0 - \boldsymbol{z}_k \rangle$$

*with* $a_0 = \frac{\alpha_0(L_0^2\alpha_0^2 - 1)}{2}$, $b_0 = 0$, $b_1 = 1$,

$$a_k = \frac{b_k(1 - \beta_k)}{2\beta_k}(\alpha_k + 2\rho_k) - b_k\rho_k \quad \text{and} \quad b_{k+1} = \frac{b_k}{1 - \beta_k}$$

*for all $k \geq 1$ satisfies $V_k \leq V_{k-1}$ for all $k \geq 1$.*

Based on the above potential lemma, we next provide a convergence analysis of FEG. The analyses for the convergence rate of FEG-A and S-FEG, *i.e.*, the proofs of Theorem 5.1 and Theorem 6.1, are similar to that of FEG and are provided in Appendix C.3 and Appendix D.3.

## 7.1 Convergence analysis for FEG

*Proof of Theorem 4.1.* Recall that FEG is equivalent to (Class FEG) with $\alpha_k = \frac{1}{L}$, $\beta_k = \frac{1}{k+1}$, and $\rho_k = \rho$. It is straightforward to verify that the given $\{\alpha_k\}_{k\geq 0}$ and $\{\beta_k\}_{k\geq 0}$ satisfy the conditions in Lemma 7.1 with $L_k = L$ for all $k \geq 0$. Since

$$a_k = \frac{b_k(1 - \beta_k)}{2\beta_k}(\alpha_k + 2\rho_k) - b_k\rho_k = \frac{k^2}{2}\left(\frac{1}{L} + 2\rho\right) - k\rho \qquad \text{and}$$

$$b_k = \frac{1}{1 - \beta_{k-1}}b_{k-1} = \left(\prod_{i=1}^{k-1}\frac{1}{1 - \beta_i}\right)b_1 = k,$$

Lemma 7.1 implies that

$$0 = V_0 \geq V_k = \left(\frac{k^2}{2}\left(\frac{1}{L} + 2\rho\right) - k\rho\right)\|\boldsymbol{F}\boldsymbol{z}_k\|^2 - k\langle \boldsymbol{F}\boldsymbol{z}_k, \boldsymbol{z}_0 - \boldsymbol{z}_k \rangle.$$

Therefore,

$$\frac{k^2}{2}\left(\frac{1}{L} + 2\rho\right)\|\boldsymbol{F}\boldsymbol{z}_k\|^2 \leq k\langle \boldsymbol{F}\boldsymbol{z}_k, \boldsymbol{z}_0 - \boldsymbol{z}_k \rangle + k\rho\|\boldsymbol{F}\boldsymbol{z}_k\|^2$$
$$= k\langle \boldsymbol{F}\boldsymbol{z}_k, \boldsymbol{z}_0 - \boldsymbol{z}_* \rangle + k\langle \boldsymbol{F}\boldsymbol{z}_k, \boldsymbol{z}_* - \boldsymbol{z}_k \rangle + k\rho\|\boldsymbol{F}\boldsymbol{z}_k\|^2$$
$$\leq k\langle \boldsymbol{F}\boldsymbol{z}_k, \boldsymbol{z}_0 - \boldsymbol{z}_* \rangle \qquad (\because \rho\text{-comonotonicity of } \boldsymbol{F})$$
$$\leq k\|\boldsymbol{F}\boldsymbol{z}_k\|\|\boldsymbol{z}_0 - \boldsymbol{z}_*\|.$$

The desired result follows directly by dividing both sides by $\frac{k^2}{2}\left(\frac{1}{L} + 2\rho\right)\|\boldsymbol{F}\boldsymbol{z}_k\|$. $\qquad\qquad\square$

## 8 Discussion: first-order methods for Lipschitz continuous operators

Throughout this paper, we studied and constructed efficient methods in a class of first-order methods:

$$\boldsymbol{z}_k \in \boldsymbol{z}_0 + \text{span}\{\boldsymbol{F}\boldsymbol{z}_0, \cdots, \boldsymbol{F}\boldsymbol{z}_k\}$$

denoted by $\mathcal{A}$, for smooth structured nonconvex-nonconcave problems. We observed that all existing first-order methods, including the FEG, required an additional condition, such as the negative comonoticity, on a Lipschitz continuous $\boldsymbol{F}$ to guarantee convergence. One would then be curious whether or not there exists an (efficient) method in class $\mathcal{A}$ that guarantees convergence without any additional condition on a Lipschitz continuous $\boldsymbol{F}$. Unfortunately, the following lemma states that there exists a *worst-case*[7] smooth example that none of the methods in $\mathcal{A}$ can find its stationary point. The corresponding smooth function is illustrated in Figure 3.

---

[7][15, 20] also introduce worst-case minimax examples that existing methods cannot find a stationary point. A key difference from our example is that their saddle-gradient operators are not Lipschitz continuous. In addition, the considered classes of methods in [15, 20] exclude EG+ and FEG, unlike the class $\mathcal{A}$.

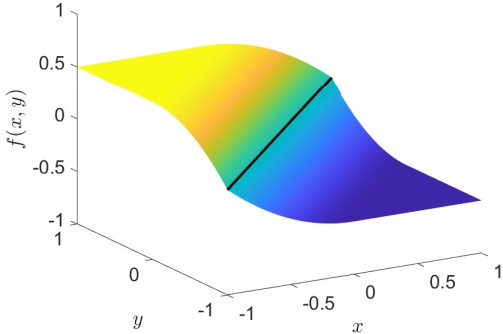

Figure 3: A smooth worst-case example $f(x, y)$ (5) with $L = R = 1$ for first-order methods. any sequence $\{z_k\}_{k \geq 0}$ generated by a first-order method in class $\mathcal{A}$ starting from $(0, 0)$ is contained in the line $x = y$.

**Lemma 8.1.** *Let us consider the following function $f : \mathbb{R}^2 \to \mathbb{R}$ for some $L, R > 0$:*

$$
f(x, y) = \begin{cases}
\frac{R}{2} & \text{for } x < y - \sqrt{\frac{R}{L}} \\
-\frac{L}{2}(x - y)^2 - \sqrt{LR}(x - y) & \text{for } y - \sqrt{\frac{R}{L}} \leq x < y \\
\frac{L}{2}(x - y)^2 - \sqrt{LR}(x - y) & \text{for } y \leq x < y + \sqrt{\frac{R}{L}} \\
-\frac{R}{2} & \text{for } y + \sqrt{\frac{R}{L}} < x.
\end{cases}
\tag{5}
$$

*Its saddle-gradient operator $\mathbf{F}$ is L-Lipschitz continuous but not comonotone.[8] Then, the sequence $\{z_k\}_{k \geq 0}$ generated by any first-order method in class $\mathcal{A}$ with $z_0 = (0, 0)$ satisfies $\|\mathbf{F}z_k\|^2 = 2LR$ for all $k \geq 0$.*

*Proof.* $\mathbf{F}$ satisfies $\mathbf{F}(x, y) = (-\sqrt{LR}, -\sqrt{LR})$ whenever $x = y$. Hence, for all sequences $\{z_k\}_{k \geq 0}$ satisfying $z_0 = (0, 0)$ and $z_k \in z_0 + \text{span}\{\mathbf{F}z_0, \cdots, \mathbf{F}z_k\}$ for all $k \geq 0$, we have that $\{z_k\}_{k \geq 0} \subseteq \{z = (x, y) \in \mathbb{R}^2 | x = y\}$; thus, $\|\mathbf{F}z_k\|^2 = 2LR$ for all $k \geq 0$. $\square$

The lemma implies that one should consider a class of methods, other than the class $\mathcal{A}$, to guarantee finding a stationary point of any smooth problem, which we leave as future work. We also leave finding additional conditions for a Lipschitz continuous $\mathbf{F}$, weaker than the weak MVI condition and the negative comonotonicity (with $\rho > -\frac{1}{2L}$), which guarantee convergence or its accelerated rate, respectively, as future work.

## 9  Conclusion

This paper proposed a two-time-scale and anchored extragradient method, named FEG, for smooth structured nonconvex-nonconcave problems. The proposed FEG has an accelerated $\mathcal{O}(1/k^2)$ rate, with respect to the squared gradient norm, for the Lipschitz continuous and negative comonotone operators for the first time. The FEG also has value for smooth convex-concave problems, compared to existing works. We further studied its backtracking line-search version, named FEG-A, for the smooth structured nonconvex-nonconcave problems and studied its stochastic version, named S-FEG, for smooth convex-concave problems. We leave extending this work to stochastic, composite, or more general nonconvex-nonconcave setting and applying to more realistic problems as future work.

---

[8] Let $z = \left(x, x + \sqrt{\frac{R}{L}}\right)$ and $w = (0, 0)$. Since $\mathbf{F}z = (0, 0)$ and $\mathbf{F}w = (-\sqrt{LR}, -\sqrt{LR})$, we get $\langle \mathbf{F}z - \mathbf{F}w, z - w \rangle = 2\sqrt{LR}x + R$ and $\|\mathbf{F}z - \mathbf{F}w\|^2 = 2LR$, which implies that $\rho = -\infty$ in the comonotonicity condition as $x \to -\infty$.

## Acknowledgments and Disclosure of Funding

This work was supported in part by the National Research Foundation of Korea (NRF) grant funded by the Korea government (MSIT) (No. 2019R1A5A1028324), the POSCO Science Fellowship of POSCO TJ Park Foundation, and the Samsung Science and Technology Foundation (No. SSTF-BA2101-02).

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
