# Appendix

## A  Proof for Section 3

### A.1  Proof of Example 1

Let $f_\eta$ be the saddle envelope of $f$ [11]:

$$f_\eta(\bar{x}, \bar{y}) := \min_{x \in \mathcal{X}} \max_{y \in \mathcal{Y}} f(x, y) + \frac{\eta}{2}\|x - \bar{x}\|^2 - \frac{\eta}{2}\|y - \bar{y}\|^2,$$

and $F_\eta$ be its saddle gradient operator. Proposition 2.10 in [11] shows that $f_\eta$ satisfies

$$\frac{\eta\alpha}{\eta + \alpha}I \preceq \nabla^2_{xx} f_\eta \preceq \eta I \quad \text{and} \quad \frac{\eta\alpha}{\eta + \alpha}I \preceq -\nabla^2_{yy} f_\eta \preceq \eta I.$$

This implies that $F_\eta$ is $\frac{\eta\alpha}{\eta+\alpha}$-strongly monotone (and thus monotone).

It is enough to show that $F_\eta$ is monotone if and only if $F$ is $-\frac{1}{\eta}$-comonotone. By Lemma 2.5 in [11], we have the relationship $F_\eta z = FRz$, where $R := \left(I + \frac{1}{\eta}F\right)^{-1}$ denotes the standard resolvent of $\frac{1}{\eta}F$. The resolvent $R$ is injective for $\eta > \gamma$. Let $Z := \mathcal{X} \times \mathcal{Y}$. Then, $F_\eta$ is monotone if and only if

$$\langle FRz - FRw, \ z - w \rangle \geq 0, \qquad \forall z, w \in Z,$$

$$\Leftrightarrow \quad \left\langle Fz' - Fw', \ \left(I + \frac{1}{\eta}F\right)z' - \left(I + \frac{1}{\eta}F\right)w' \right\rangle \geq 0, \qquad \forall z, w \in Z, z' = Rz, w' = Rw,$$

$$\Leftrightarrow \quad \langle Fz' - Fw', \ z' - w' \rangle \geq -\frac{1}{\eta}\|Fz' - Fw'\|^2, \qquad \forall z', w' \in Z,$$

which corresponds to the $-\frac{1}{\eta}$-comonotonicity of $F$. $\qquad\square$

## B  Proof for Section 4

### B.1  Proof of Example 2

Starting from $z_0 = (1, 0)$, it is easy to verify that $z_{1/2} = (1, 0)$, $z_1 = (1, 1)$, $z_{1+1/2} = \left(\frac{1}{2}, 1\right)$, and $z_2 = (0, 1)$. We next use the induction to show that $z_k = \left(0, \frac{2}{k}\right)$ for $k = 4l + 2$ and for all $l = 0, 1, 2, \dots$. Assume that $z_k = \left(0, \frac{2}{k}\right)$ for some $k = 4l + 2$. Then, the next eight consecutive iterates are as follows:

$$z_{k+1/2} = z_k + \frac{1}{k+1}(z_0 - z_k) - \left(1 - \frac{1}{k+1}\right)\frac{1}{L}Fz_k$$

$$= \left(\frac{1}{k+1}, \frac{2}{k+1}\right) - \frac{k}{k+1}\left(\frac{2}{k}, 0\right) = \left(-\frac{1}{k+1}, \frac{2}{k+1}\right),$$

$$z_{k+1} = z_k + \frac{1}{k+1}(z_0 - z_k) - \frac{1}{L}Fz_{k+1/2}$$

$$= \left(\frac{1}{k+1}, \frac{2}{k+1}\right) - \left(\frac{2}{k+1}, \frac{1}{k+1}\right) = \left(-\frac{1}{k+1}, \frac{1}{k+1}\right),$$

$$z_{k+1+1/2} = z_{k+1} + \frac{1}{k+2}(z_0 - z_{k+1}) - \left(1 - \frac{1}{k+2}\right)\frac{1}{L}Fz_{k+1}$$

$$= \left(0, \frac{1}{k+2}\right) - \frac{k+1}{k+2}\left(\frac{1}{k+1}, \frac{1}{k+1}\right) = \left(-\frac{1}{k+2}, 0\right),$$

$$z_{k+2} = z_{k+1} + \frac{1}{k+2}(z_0 - z_{k+1}) - \frac{1}{L}Fz_{k+1+1/2}$$

$$= \left(0, \frac{1}{k+2}\right) - \left(0, \frac{1}{k+2}\right) = (0, 0)$$

$$z_{k+2+1/2} = z_{k+2} + \frac{1}{k+3}(z_0 - z_{k+2}) - \left(1 - \frac{1}{k+3}\right)\frac{1}{L}Fz_{k+2}$$

$$= \left(\frac{1}{k+3}, 0\right),$$

$$z_{k+3} = z_{k+2} + \frac{1}{k+3}(z_0 - z_{k+2}) - \frac{1}{L}\boldsymbol{F}z_{k+2+1/2}$$
$$= \left(\frac{1}{k+3}, 0\right) - \left(0, -\frac{1}{k+3}\right) = \left(\frac{1}{k+3}, \frac{1}{k+3}\right),$$
$$z_{k+3+1/2} = z_{k+3} + \frac{1}{k+4}(z_0 - z_{k+3}) - \left(1 - \frac{1}{k+4}\right)\frac{1}{L}\boldsymbol{F}z_{k+3}$$
$$= \left(\frac{2}{k+4}, \frac{1}{k+4}\right) - \frac{k+3}{k+4}\left(\frac{1}{k+3}, -\frac{1}{k+3}\right) = \left(\frac{1}{k+4}, \frac{2}{k+4}\right),$$
$$z_{k+4} = z_{k+3} + \frac{1}{k+4}(z_0 - z_{k+3}) - \frac{1}{L}\boldsymbol{F}z_{k+3+1/2}$$
$$= \left(\frac{2}{k+4}, \frac{1}{k+4}\right) - \left(\frac{2}{k+4}, -\frac{1}{k+4}\right) = \left(0, \frac{2}{k+4}\right),$$

so $z_{4l+6} = \left(0, \frac{2}{4l+6}\right)$. Therefore, we get $z_{4l+2} = \left(0, \frac{1}{2l+1}\right)$ for all $l \geq 0$. $\qquad\square$

## C Algorithm and proofs for Section 5

### C.1 A detailed description of FEG-A

A detailed description of the FEG-A, in Algorithm 2, is provided in Algorithm 4.

### C.2 Proof of Lemma 5.1

We show that $\tau_k \geq \underline{\tau} := \min\left\{\tau_{-1}, \frac{1-\delta}{L}\right\}$ for all $k \geq 0$, and $\eta_k \geq \min\{\eta_0, (1-\delta)(\tau_k + 2\rho)\}$ for all $k \geq 1$ by contradiction. Note that since $\tau_{-1} > \max\{0, -2\rho\}$ and $\rho > -\frac{1-\delta}{2L}$, both $\underline{\tau}$ and $\underline{\eta}$ are positive.

First, suppose that $\tau_k < \underline{\tau}$ for some $k \geq 0$. (1) For the case $\tau_{-1} \leq \frac{1-\delta}{L}$, we get $\tau_k = \tau_{-1}$ for all $k \geq 0$ by the definition of $\tau_k$, which contradicts to the assumption $\tau_k < \tau_{-1}$. (2) Consider the case $\tau_{-1} > \frac{1-\delta}{L}$, where the assumption reduces to $\tau_k < \frac{1-\delta}{L}$. For $k = 0$, by the definition of $\tau_0$, we get $\|\boldsymbol{F}\hat{z}_1 - \boldsymbol{F}z_0\| > \frac{1-\delta}{\tau_0}\|\hat{z}_1 - z_0\|$ where $\hat{z}_1 = z_0 - \frac{\tau_0}{1-\delta}\boldsymbol{F}z_0$, which contradicts to the $L$-Lipschitz

---

**Algorithm 4** Fast extragradient method with adaptive step size (FEG-A)

---

**Input:** $z_0 \in \mathbb{R}^d$, $\tau_{-1} \in (\max\{0, -2\rho\}, \infty)$, $\eta_0 \in (0, \infty)$, $\delta \in (0, 1)$
Find the smallest nonnegative integer $i_0$ such that $\hat{z} = z_0 - \tau_{-1}(1-\delta)^{i_0}\boldsymbol{F}z_0$ satisfies $\|\boldsymbol{F}\hat{z} - \boldsymbol{F}z_0\| \leq \frac{1}{\tau_{-1}(1-\delta)^{i_0}}\|\hat{z} - z_0\|$.
$\tau_0 = \tau_{-1}(1-\delta)^{i_0}$, $z_1 = z_0 - \tau_0\boldsymbol{F}z_0$.
**for** $k = 1, 2, \ldots$ **do**
    $i_k = j_k = 0$, searching = True
    **while** searching = True **do**
        searching = False, $\tau_k = \tau_{k-1}(1-\delta)^{i_k}$, $\eta_k = \eta_{k-1}(1-\delta)^{j_k}$

$$z_{k+1/2} = z_k + \frac{1}{k+1}(z_0 - z_k) - \left(1 - \frac{1}{k+1}\right)\eta_k\boldsymbol{F}z_k \quad \text{and}$$
$$z_{k+1} = z_k + \frac{1}{k+1}(z_0 - z_k) - \tau_k\boldsymbol{F}z_{k+1/2} - \left(1 - \frac{1}{k+1}\right)(\eta_k - \tau_k)\boldsymbol{F}z_k$$

        **if** $\|\boldsymbol{F}z_{k+1} - \boldsymbol{F}z_{k+1/2}\| > \frac{1}{\tau_k}\|z_{k+1} - z_{k+1/2}\|$ **then**
            $i_k \leftarrow i_k + 1$
            searching = True
        **end if**
        **if** $\langle \boldsymbol{F}z_{k+1} - \boldsymbol{F}z_k, z_{k+1} - z_k\rangle < \frac{\eta_k - \tau_k}{2}\|\boldsymbol{F}z_{k+1} - \boldsymbol{F}z_k\|^2$ **then**
            $j_k \leftarrow j_k + 1$
            searching = True
        **end if**
    **end while**
**end for**

---

continuity of $\boldsymbol{F}$ as $\frac{1-\delta}{\tau_0} > L$. For $k \geq 1$, by the definition of $\tau_k$, there exists $i \leq k$ such that the two corresponding iterates

$$\hat{z}_{i+1/2} = z_i + \frac{1}{i+1}(z_0 - z_i) - \left(1 - \frac{1}{i+1}\right)\hat{\eta}_i \boldsymbol{F} z_i \quad \text{and}$$

$$\hat{z}_{i+1} = z_i + \frac{1}{i+1}(z_0 - z_i) - \frac{\tau_k}{1-\delta}\boldsymbol{F}\hat{z}_{i+1/2} - \left(1 - \frac{1}{i+1}\right)\left(\hat{\eta}_i - \frac{\tau_k}{1-\delta}\right)\boldsymbol{F} z_i$$

satisfy $\|\boldsymbol{F}\hat{z}_{i+1} - \boldsymbol{F}\hat{z}_{i+1/2}\| > \frac{1-\delta}{\tau_k}\|\hat{z}_{i+1} - \hat{z}_{i+1/2}\|$ for some $\hat{\eta}_i > 0$. However, this inequality contradicts to the $L$-Lipschitz continuity of $\boldsymbol{F}$ as $\frac{1-\delta}{\tau_k} > L$. Therefore, we have $\tau_k \geq \underline{\tau} > 0$ for all $k \geq 0$.

Similarly, suppose that $\eta_k < \min\{\eta_0, (1-\delta)(\tau_k + 2\rho)\}$ for some $k \geq 1$. (1) For the case $\eta_0 \leq (1-\delta)(\tau_k + 2\rho)$, we get $\eta_i = \eta_0$ for all $1 \leq i \leq k$ by the definition of $\eta_k$, which contradicts to the assumption $\eta_k < \eta_0$. (2) Consider the case $\eta_0 > (1-\delta)(\tau_k + 2\rho)$, where the assumption reduces to $\eta_k < (1-\delta)(\tau_k + 2\rho)$. Then by the definition of $\eta_k$, there exists $i \leq k$ such that the two corresponding iterates

$$\hat{z}_{i+1/2} = z_i + \frac{1}{i+1}(z_0 - z_i) - \left(1 - \frac{1}{i+1}\right)\frac{\eta_k}{1-\delta}\boldsymbol{F} z_i \quad \text{and}$$

$$\hat{z}_{i+1} = z_i + \frac{1}{i+1}(z_0 - z_i) - \hat{\tau}_i \boldsymbol{F}\hat{z}_{i+1/2} - \left(1 - \frac{1}{i+1}\right)\left(\frac{\eta_k}{1-\delta} - \hat{\tau}_i\right)\boldsymbol{F} z_i$$

satisfy $\langle \boldsymbol{F}\hat{z}_{i+1} - \boldsymbol{F}\hat{z}_i, \hat{z}_{i+1} - \hat{z}_i \rangle < \frac{\frac{\eta_k}{1-\delta} - \hat{\tau}_i}{2}\|\boldsymbol{F}\hat{z}_{i+1} - \boldsymbol{F}\hat{z}_i\|^2$ for some $\hat{\tau}_i \geq \tau_i$. However, this inequality contradicts to the $\rho$-comonotonicity of $\boldsymbol{F}$ as $\frac{\frac{\eta_k}{1-\delta} - \hat{\tau}_i}{2} < \frac{\tau_k + 2\rho - \hat{\tau}_i}{2} \leq \rho$. Therefore, we have $\eta_k \geq \min\{\eta_0, (1-\delta)(\tau_k + 2\rho)\}$ for all $k \geq 0$. $\qquad\square$

### C.3  Proof of Theorem 5.1

Note that FEG-A is equivalent to (Class FEG) with $\alpha_k = \tau_k$, $\beta_k = \frac{1}{k+1}$, and $\rho_k = \frac{\eta_k - \tau_k}{2}$. The given sequence in FEG-A satisfies the conditions in Lemma 7.1 with $L_k = \frac{1}{\tau_k}$:

$$\frac{(1-\beta_{k+1})}{2\beta_{k+1}}(\alpha_{k+1} + 2\rho_{k+1}) - \rho_{k+1} = \frac{k}{2}\eta_{k+1} + \frac{1}{2}\tau_{k+1} \leq \frac{k}{2}\eta_k + \frac{1}{2}\tau_k = \frac{1}{2\beta_k}(\alpha_k + 2\rho_k) - \rho_k$$

where the inequality follows from the fact that $\{\tau_k\}_{k \geq 0}$ and $\{\eta_k\}_{k \geq 0}$ are nonincreasing sequences. Since

$$a_k = \frac{b_k(1-\beta_k)}{2\beta_k}(\alpha_k + 2\rho_k) - b_k\rho_k = \frac{k}{2}((k-1)\eta_k + \tau_k) \quad \text{and}$$

$$b_k = \frac{1}{1-\beta_{k-1}}b_{k-1} = \left(\prod_{i=1}^{k-1}\frac{1}{1-\beta_i}\right)b_1 = k,$$

Lemma 7.1 implies that

$$0 = V_0 \geq V_k = \frac{k}{2}((k-1)\eta_k + \tau_k)\|\boldsymbol{F} z_k\|^2 - k\langle \boldsymbol{F} z_k, z_0 - z_k \rangle.$$

Therefore,

$$\frac{k}{2}((k-1)\eta_k + \tau_k + 2\rho)\|\boldsymbol{F} z_k\|^2 \leq k\langle \boldsymbol{F} z_k, z_0 - z_k \rangle + k\rho\|\boldsymbol{F} z_k\|^2$$

$$= k\langle \boldsymbol{F} z_k, z_0 - z_* \rangle + k\langle \boldsymbol{F} z_k, z_* - z_k \rangle + k\rho\|\boldsymbol{F} z_k\|^2$$

$$\leq k\langle \boldsymbol{F} z_k, z_0 - z_* \rangle \quad (\because \rho\text{-comonotonicity of } \boldsymbol{F})$$

$$\leq k\|\boldsymbol{F} z_k\|\|z_0 - z_*\|.$$

Then by dividing both sides by $\frac{k}{2}((k-1)\eta_k + \tau_k + 2\rho)\|\boldsymbol{F} z_k\|$ and using Lemma 5.1, we get

$$\|\boldsymbol{F} z_k\| \leq \frac{2\|z_0 - z_*\|}{(k-1)\eta_k + \tau_k + 2\rho}.$$

$\qquad\square$

## D  Proofs for Section 7

### D.1  Proof of Lemma 7.1

First, for $k = 0$, note that

$$
\begin{aligned}
V_1 &= a_1\|\boldsymbol{F}\boldsymbol{z}_1\|^2 - b_1\langle \boldsymbol{F}\boldsymbol{z}_1,\, \boldsymbol{z}_0 - \boldsymbol{z}_1\rangle \\
&= a_1\|\boldsymbol{F}\boldsymbol{z}_1\|^2 - \alpha_0 b_1\langle \boldsymbol{F}\boldsymbol{z}_1,\, \boldsymbol{F}\boldsymbol{z}_0\rangle \\
&= \Big(\frac{b_1(1-\beta_1)}{2\beta_1}(\alpha_1 + 2\rho_1) - b_1\rho_1\Big)\|\boldsymbol{F}\boldsymbol{z}_1\|^2 - \alpha_0\langle \boldsymbol{F}\boldsymbol{z}_1,\, \boldsymbol{F}\boldsymbol{z}_0\rangle \\
&\leq \Big(\frac{b_1}{2\beta_0}(\alpha_0 + 2\rho_0) - b_1\rho_0\Big)\|\boldsymbol{F}\boldsymbol{z}_1\|^2 - \alpha_0\langle \boldsymbol{F}\boldsymbol{z}_1,\, \boldsymbol{F}\boldsymbol{z}_0\rangle \\
&= \frac{\alpha_0}{2}\|\boldsymbol{F}\boldsymbol{z}_1\|^2 - \alpha_0\langle \boldsymbol{F}\boldsymbol{z}_1,\, \boldsymbol{F}\boldsymbol{z}_0\rangle. \tag{6}
\end{aligned}
$$

By the given condition, we get

$$
0 \leq L_0^2\|\boldsymbol{z}_1 - \boldsymbol{z}_0\|^2 - \|\boldsymbol{F}\boldsymbol{z}_1 - \boldsymbol{F}\boldsymbol{z}_0\|^2 = L_0^2\alpha_0^2\|\boldsymbol{F}\boldsymbol{z}_0\|^2 - \|\boldsymbol{F}\boldsymbol{z}_1 - \boldsymbol{F}\boldsymbol{z}_0\|^2. \tag{7}
$$

Hence, the sum of (6) and (7) with multiplying factor $\frac{\alpha_0}{2}$ yields

$$
\begin{aligned}
V_1 &\leq \frac{\alpha_0}{2}\|\boldsymbol{F}\boldsymbol{z}_1\|^2 - \alpha_0\langle \boldsymbol{F}\boldsymbol{z}_1,\, \boldsymbol{F}\boldsymbol{z}_0\rangle + \frac{\alpha_0}{2}(L_0^2\alpha_0^2\|\boldsymbol{F}\boldsymbol{z}_0\|^2 - \|\boldsymbol{F}\boldsymbol{z}_1 - \boldsymbol{F}\boldsymbol{z}_0\|^2) \\
&= \frac{\alpha_0(L_0^2\alpha_0^2 - 1)}{2}\|\boldsymbol{F}\boldsymbol{z}_0\|^2 = V_0.
\end{aligned}
$$

Next, for $k \geq 1$, here we note the following relations for later use:

$$
\begin{aligned}
\boldsymbol{z}_{k+1} - \boldsymbol{z}_k &= \frac{\beta_k}{1-\beta_k}(\boldsymbol{z}_0 - \boldsymbol{z}_{k+1}) - \frac{\alpha_k}{1-\beta_k}\boldsymbol{F}\boldsymbol{z}_{k+1/2} - 2\rho_k\boldsymbol{F}\boldsymbol{z}_k, \\
\boldsymbol{z}_{k+1} - \boldsymbol{z}_k &= \beta_k(\boldsymbol{z}_0 - \boldsymbol{z}_k) - \alpha_k\boldsymbol{F}\boldsymbol{z}_{k+1/2} - 2(1-\beta_k)\rho_k\boldsymbol{F}\boldsymbol{z}_k, \text{ and} \\
\boldsymbol{z}_{k+1} - \boldsymbol{z}_{k+1/2} &= \alpha_k((1-\beta_k)\boldsymbol{F}\boldsymbol{z}_k - \boldsymbol{F}\boldsymbol{z}_{k+1/2}).
\end{aligned}
$$

Then, by the given condition, we have

$$
\begin{aligned}
V_k - V_{k+1} \geq\, & V_k - V_{k+1} - \frac{b_k}{\beta_k}\big(\langle \boldsymbol{F}\boldsymbol{z}_{k+1} - \boldsymbol{F}\boldsymbol{z}_k,\, \boldsymbol{z}_{k+1} - \boldsymbol{z}_k\rangle - \rho_k\|\boldsymbol{F}\boldsymbol{z}_{k+1} - \boldsymbol{F}\boldsymbol{z}_k\|^2\big) \\
=\, & V_k - V_{k+1} - \frac{b_k}{\beta_k}\langle \boldsymbol{F}\boldsymbol{z}_{k+1},\, \boldsymbol{z}_{k+1} - \boldsymbol{z}_k\rangle + \frac{b_k}{\beta_k}\langle \boldsymbol{F}\boldsymbol{z}_k,\, \boldsymbol{z}_{k+1} - \boldsymbol{z}_k\rangle + \frac{b_k\rho_k}{\beta_k}\|\boldsymbol{F}\boldsymbol{z}_{k+1} - \boldsymbol{F}\boldsymbol{z}_k\|^2 \\
=\, & (a_k\|\boldsymbol{F}\boldsymbol{z}_k\|^2 - b_k\langle \boldsymbol{F}\boldsymbol{z}_k,\, \boldsymbol{z}_0 - \boldsymbol{z}_k\rangle) - (a_{k+1}\|\boldsymbol{F}\boldsymbol{z}_{k+1}\|^2 - b_{k+1}\langle \boldsymbol{F}\boldsymbol{z}_{k+1},\, \boldsymbol{z}_0 - \boldsymbol{z}_{k+1}\rangle) \\
& - \frac{b_k}{\beta_k}\Big\langle \boldsymbol{F}\boldsymbol{z}_{k+1},\, \frac{\beta_k}{1-\beta_k}(\boldsymbol{z}_0 - \boldsymbol{z}_{k+1}) - \frac{\alpha_k}{1-\beta_k}\boldsymbol{F}\boldsymbol{z}_{k+1/2} - 2\rho_k\boldsymbol{F}\boldsymbol{z}_k \Big\rangle \\
& + \frac{b_k}{\beta_k}\langle \boldsymbol{F}\boldsymbol{z}_k,\, \beta_k(\boldsymbol{z}_0 - \boldsymbol{z}_k) - \alpha_k\boldsymbol{F}\boldsymbol{z}_{k+1/2} - 2(1-\beta_k)\rho_k\boldsymbol{F}\boldsymbol{z}_k\rangle \\
& + \frac{b_k\rho_k}{\beta_k}\|\boldsymbol{F}\boldsymbol{z}_{k+1} - \boldsymbol{F}\boldsymbol{z}_k\|^2 \\
=\, & \Big(a_k - \frac{b_k(1-2\beta_k)\rho_k}{\beta_k}\Big)\|\boldsymbol{F}\boldsymbol{z}_k\|^2 - \Big(-\frac{b_k\rho_k}{\beta_k} + a_{k+1}\Big)\|\boldsymbol{F}\boldsymbol{z}_{k+1}\|^2 \\
& + \Big(b_{k+1} - \frac{b_k}{1-\beta_k}\Big)\langle \boldsymbol{F}\boldsymbol{z}_{k+1},\, \boldsymbol{z}_0 - \boldsymbol{z}_{k+1}\rangle + \frac{b_k\alpha_k}{\beta_k(1-\beta_k)}\langle \boldsymbol{F}\boldsymbol{z}_{k+1},\, \boldsymbol{F}\boldsymbol{z}_{k+1/2}\rangle \\
& - \frac{\alpha_k b_k}{\beta_k}\langle \boldsymbol{F}\boldsymbol{z}_k,\, \boldsymbol{F}\boldsymbol{z}_{k+1/2}\rangle \\
=\, & \Big(a_k - \frac{b_k(1-2\beta_k)\rho_k}{\beta_k}\Big)\|\boldsymbol{F}\boldsymbol{z}_k\|^2 - \Big(-\frac{b_k\rho_k}{\beta_k} + a_{k+1}\Big)\|\boldsymbol{F}\boldsymbol{z}_{k+1}\|^2 \\
& + \frac{b_k\alpha_k}{\beta_k(1-\beta_k)}\langle \boldsymbol{F}\boldsymbol{z}_{k+1},\, \boldsymbol{F}\boldsymbol{z}_{k+1/2}\rangle - \frac{\alpha_k b_k}{\beta_k}\langle \boldsymbol{F}\boldsymbol{z}_k,\, \boldsymbol{F}\boldsymbol{z}_{k+1/2}\rangle. \qquad \Big(\because b_{k+1} = \frac{b_k}{1-\beta_k}.\Big)
\end{aligned}
$$

$$\tag{8}$$

By the given condition, we also have

$$0 \geq \|\boldsymbol{F}\boldsymbol{z}_{k+1} - \boldsymbol{F}\boldsymbol{z}_{k+1/2}\|^2 - L_k^2 \|\boldsymbol{z}_{k+1} - \boldsymbol{z}_{k+1/2}\|^2$$

$$= \|\boldsymbol{F}\boldsymbol{z}_{k+1} - \boldsymbol{F}\boldsymbol{z}_{k+1/2}\|^2 - L_k^2 \alpha_k^2 \|(1-\beta_k)\boldsymbol{F}\boldsymbol{z}_k - \boldsymbol{F}\boldsymbol{z}_{k+1/2}\|^2. \tag{9}$$

Hence, the sum of (8) and (9) with multiplying factor $\frac{b_k}{2L_k^2 \alpha_k \beta_k (1-\beta_k)}$ yields

$$V_k - V_{k+1} \geq \left( a_k - \frac{b_k(1-2\beta_k)\rho_k}{\beta_k} \right) \|\boldsymbol{F}\boldsymbol{z}_k\|^2 - \left( -\frac{b_k \rho_k}{\beta_k} + a_{k+1} \right) \|\boldsymbol{F}\boldsymbol{z}_{k+1}\|^2$$

$$+ \frac{b_k \alpha_k}{\beta_k(1-\beta_k)} \langle \boldsymbol{F}\boldsymbol{z}_{k+1}, \boldsymbol{F}\boldsymbol{z}_{k+1/2} \rangle - \frac{\alpha_k b_k}{\beta_k} \langle \boldsymbol{F}\boldsymbol{z}_k, \boldsymbol{F}\boldsymbol{z}_{k+1/2} \rangle$$

$$+ \frac{b_k}{2L_k^2 \alpha_k \beta_k (1-\beta_k)} (\|\boldsymbol{F}\boldsymbol{z}_{k+1} - \boldsymbol{F}\boldsymbol{z}_{k+1/2}\|^2 - L_k^2 \alpha_k^2 \|(1-\beta_k)\boldsymbol{F}\boldsymbol{z}_k - \boldsymbol{F}\boldsymbol{z}_{k+1/2}\|^2)$$

$$= \left( a_k - \frac{b_k(1-2\beta_k)\rho_k}{\beta_k} - \frac{b_k(1-\beta_k)\alpha_k}{2\beta_k} \right) \|\boldsymbol{F}\boldsymbol{z}_k\|^2 + \left( \frac{b_k}{2L_k^2 \alpha_k \beta_k (1-\beta_k)} + \frac{b_k \rho_k}{\beta_k} - a_{k+1} \right) \|\boldsymbol{F}\boldsymbol{z}_{k+1}\|^2$$

$$- \left( \frac{b_k}{L_k^2 \alpha_k \beta_k (1-\beta_k)} - \frac{b_k \alpha_k}{\beta_k(1-\beta_k)} \right) \langle \boldsymbol{F}\boldsymbol{z}_{k+1}, \boldsymbol{F}\boldsymbol{z}_{k+1/2} \rangle$$

$$+ \frac{b_k}{2L_k^2 \alpha_k \beta_k (1-\beta_k)} (1 - L_k^2 \alpha_k^2) \|\boldsymbol{F}\boldsymbol{z}_{k+1/2}\|^2$$

$$= \left( \frac{b_k}{2L_k^2 \alpha_k (1-\beta_k)\beta_k} + \frac{b_k \rho_k}{\beta_k} - a_{k+1} \right) \|\boldsymbol{F}\boldsymbol{z}_{k+1}\|^2$$

$$- \frac{b_k(1 - L_k^2 \alpha_k^2)}{2L_k^2 \alpha_k \beta_k (1-\beta_k)} 2 \langle \boldsymbol{F}\boldsymbol{z}_{k+1}, \boldsymbol{F}\boldsymbol{z}_{k+1/2} \rangle$$

$$+ \frac{b_k(1 - L_k^2 \alpha_k^2)}{2L_k^2 \alpha_k \beta_k (1-\beta_k)} \|\boldsymbol{F}\boldsymbol{z}_{k+1/2}\|^2.$$

$$\left( \because a_k = \frac{b_k(1-\beta_k)}{2\beta_k}(\alpha_k + 2\rho_k) - b_k \rho_k = \frac{b_k(1-\beta_k)\alpha_k}{2\beta_k} + \frac{b_k(1-2\beta_k)\rho_k}{\beta_k}. \right)$$

Note that the given conditions imply that

$$a_{k+1} = \frac{b_{k+1}(1-\beta_{k+1})}{2\beta_{k+1}}(\alpha_{k+1} + 2\rho_{k+1}) - b_{k+1}\rho_{k+1}$$

$$\leq \frac{b_{k+1}}{2\beta_k}(\alpha_k + 2\rho_k) - b_{k+1}\rho_k$$

$$= \frac{b_{k+1}}{2\beta_k}\alpha_k + \frac{b_{k+1}(1-\beta_k)\rho_k}{\beta_k}$$

$$= \frac{b_k}{2\beta_k(1-\beta_k)}\alpha_k + \frac{b_k \rho_k}{\beta_k}. \qquad \left( \because b_{k+1} = \frac{b_k}{1-\beta_k}. \right)$$

Therefore, we get

$$V_k - V_{k+1} \geq \frac{b_k(1 - L_k^2 \alpha_k^2)}{2L_k^2 \alpha_k \beta_k (1-\beta_k)} (\|\boldsymbol{F}\boldsymbol{z}_{k+1}\|^2 - 2 \langle \boldsymbol{F}\boldsymbol{z}_{k+1}, \boldsymbol{F}\boldsymbol{z}_{k+1/2} \rangle + \|\boldsymbol{F}\boldsymbol{z}_{k+1/2}\|^2)$$

$$= \frac{b_k(1 - L_k^2 \alpha_k^2)}{2L_k^2 \alpha_k \beta_k (1-\beta_k)} \|\boldsymbol{F}\boldsymbol{z}_{k+1} - \boldsymbol{F}\boldsymbol{z}_{k+1/2}\|^2$$

$$\geq 0.$$

Note that $\{\alpha_k\}_{k \geq 1} \subseteq \left( 0, \frac{1}{L_k} \right]$ and $\{\beta_k\}_{k \geq 1} \subseteq (0,1)$ are the sufficient conditions for $\frac{b_k}{2L_k^2 \alpha_k \beta_k (1-\beta_k)} \geq 0$ and $\frac{b_k(1-L_k^2 \alpha_k^2)}{2L_k^2 \alpha_k \beta_k (1-\beta_k)} \geq 0$ for all $k \geq 1$. □

## D.2 Convergence analysis for S-FEG

In this section, we consider the following class of stochastic methods:

$$\boldsymbol{z}_{k+1/2} = \boldsymbol{z}_k + \beta_k(\boldsymbol{z}_0 - \boldsymbol{z}_k) - (1-\beta_k)\alpha_k \tilde{\boldsymbol{F}}\boldsymbol{z}_k$$

$$\boldsymbol{z}_{k+1} = \boldsymbol{z}_k + \beta_k(\boldsymbol{z}_0 - \boldsymbol{z}_k) - \alpha_k \tilde{\boldsymbol{F}}\boldsymbol{z}_{k+1/2}.$$

(Class S-FEG)

As in the previous section, our analysis relies on the potential function, $V_k = a_k \|\boldsymbol{F}\boldsymbol{z}_k\|^2 - b_k \langle \boldsymbol{F}\boldsymbol{z}_k, \boldsymbol{z}_0 - \boldsymbol{z}_k \rangle$. Although the expectation of the potential function is no longer nonincreasing, we have a lower bound on $\mathbb{E}[V_k] - \mathbb{E}[V_{k+1}]$ that consists of $\sigma_k^2$ and $\sigma_{k+1/2}^2$ below.

**Lemma D.1.** *Let* $\{\boldsymbol{z}_k\}_{k\geq 0}$ *be the sequence generated by* (Class S-FEG) *with* $\{\alpha_k\}_{k\geq 0}$ *and* $\{\beta_k\}_{k\geq 0}$ *satisfying* $\alpha_0 \in (0,\infty)$, $\alpha_k \in \left(0, \frac{1}{L}\right]$, $\beta_0 = 1$, $\{\beta_k\}_{k\geq 1} \subseteq (0,1)$ *for all* $k \geq 1$, *and*

$$\frac{(1 - \beta_{k+1})\alpha_{k+1}}{2\beta_{k+1}} \leq \frac{\alpha_k}{2\beta_k}$$

*for all* $k \geq 0$. *Assume that* $\boldsymbol{F}$ *is L-Lipschitz continuous and monotone, and let* $\tilde{\boldsymbol{F}}\boldsymbol{z}_{k/2} = \boldsymbol{F}\boldsymbol{z}_{k/2} + \xi_{k/2}$, *where* $\{\xi_{k/2}\}_{k\geq 0}$ *are independent random variables satisfying* $\mathbb{E}[\xi_{k/2}] = 0$ *and* $\mathbb{E}[\|\xi_{k/2}\|^2] = \sigma_{k/2}^2$ *for all* $k \geq 0$. *Then* $V_k = a_k\|\boldsymbol{F}\boldsymbol{z}_k\|^2 - b_k \langle \boldsymbol{F}\boldsymbol{z}_k, \boldsymbol{z}_0 - \boldsymbol{z}_k \rangle$ *with* $a_0 = \frac{\alpha_0(L^2\alpha_0^2 - 1)}{2}$, $b_0 = 0$, $b_1 = 1$,

$$a_k = \frac{b_k(1 - \beta_k)\alpha_k}{2\beta_k} \quad and \quad b_{k+1} = \frac{b_k}{1 - \beta_k}$$

*for all* $k \geq 1$ *satisfies*

$$\mathbb{E}[V_0] - \mathbb{E}[V_1] \geq -\left(\frac{L^2\alpha_0^3}{2} + L\alpha_0^2\right)\sigma_0^2 \quad and$$

$$\mathbb{E}[V_k] - \mathbb{E}[V_{k+1}] \geq -\frac{b_k\alpha_k(1 + 2L\alpha_k)}{2\beta_k}\left((1 - \beta_k)\sigma_k^2 + \frac{1}{1 - \beta_k}\sigma_{k+1/2}^2\right)$$

*for all* $k \geq 1$.

We first prove the following lemma that is used in the proof of Lemma D.1.

**Lemma D.2.** *Let* $\tilde{\boldsymbol{F}}\boldsymbol{z}_{k/2} = \boldsymbol{F}\boldsymbol{z}_{k/2} + \xi_{k/2}$, *where* $\{\xi_{k/2}\}_{k\geq 0}$ *are independent random variables satisfying* $\mathbb{E}[\xi_{k/2}] = 0$ *and* $\mathbb{E}[\|\xi_{k/2}\|^2] = \sigma_{k/2}^2$ *for all* $k \geq 0$. *Then, for the L-Lipschitz continuous and monotone operator* $\boldsymbol{F}$, *the sequence* $\{\boldsymbol{z}_k\}_{k\geq 0}$ *generated by* (Class S-FEG) *satisfies*

$$|\mathbb{E}[\langle \boldsymbol{F}\boldsymbol{z}_1, \tilde{\boldsymbol{F}}\boldsymbol{z}_0 - \boldsymbol{F}\boldsymbol{z}_0 \rangle]| \leq L\alpha_0\sigma_0^2$$

*and, for all* $k = 0, 1, \ldots,$

$$|\mathbb{E}[\langle \boldsymbol{F}\boldsymbol{z}_{k+1/2}, \tilde{\boldsymbol{F}}\boldsymbol{z}_k - \boldsymbol{F}\boldsymbol{z}_k \rangle]| \leq L(1 - \beta_k)\alpha_k\sigma_k^2$$

$$|\mathbb{E}[\langle \boldsymbol{F}\boldsymbol{z}_{k+1}, \tilde{\boldsymbol{F}}\boldsymbol{z}_{k+1/2} - \boldsymbol{F}\boldsymbol{z}_{k+1/2} \rangle]| \leq L\alpha_k\sigma_{k+1/2}^2.$$

*Proof.* We have that

$$\begin{aligned}
|\mathbb{E}[\langle \boldsymbol{F}\boldsymbol{z}_1, \tilde{\boldsymbol{F}}\boldsymbol{z}_0 - \boldsymbol{F}\boldsymbol{z}_0 \rangle]| &= |\mathbb{E}[\langle \boldsymbol{F}\boldsymbol{z}_1 - \boldsymbol{F}(\boldsymbol{z}_0 - \alpha_0\boldsymbol{F}\boldsymbol{z}_0), \tilde{\boldsymbol{F}}\boldsymbol{z}_0 - \boldsymbol{F}\boldsymbol{z}_0 \rangle]| \\
&\leq \mathbb{E}[\|\boldsymbol{F}\boldsymbol{z}_1 - \boldsymbol{F}(\boldsymbol{z}_0 - \alpha_0\boldsymbol{F}\boldsymbol{z}_0)\| \, \|\tilde{\boldsymbol{F}}\boldsymbol{z}_0 - \boldsymbol{F}\boldsymbol{z}_0\|] \\
&\leq \mathbb{E}[L\|\boldsymbol{z}_1 - (\boldsymbol{z}_0 - \alpha_0\boldsymbol{F}\boldsymbol{z}_0)\| \, \|\tilde{\boldsymbol{F}}\boldsymbol{z}_0 - \boldsymbol{F}\boldsymbol{z}_0\|] \\
&= \mathbb{E}[L\alpha_0\|\tilde{\boldsymbol{F}}\boldsymbol{z}_0 - \boldsymbol{F}\boldsymbol{z}_0\|^2] \\
&= L\alpha_0\sigma_0^2,
\end{aligned}$$

where the first equality uses the assumption that $\xi_0 = \tilde{\boldsymbol{F}}\boldsymbol{z}_0 - \boldsymbol{F}\boldsymbol{z}_0$ is an independent random variable with $\mathbb{E}[\xi_0] = 0$. Similarly, we have that

$$\begin{aligned}
|\mathbb{E}[\langle \boldsymbol{F}\boldsymbol{z}_{k+1/2}, \tilde{\boldsymbol{F}}\boldsymbol{z}_k - \boldsymbol{F}\boldsymbol{z}_k \rangle]| &= |\mathbb{E}[\langle \boldsymbol{F}\boldsymbol{z}_{k+1/2} - \boldsymbol{F}(\boldsymbol{z}_k + \beta_k(\boldsymbol{z}_0 - \boldsymbol{z}_k) - (1 - \beta_k)\alpha_k\boldsymbol{F}\boldsymbol{z}_k), \tilde{\boldsymbol{F}}\boldsymbol{z}_k - \boldsymbol{F}\boldsymbol{z}_k \rangle]| \\
&\leq \mathbb{E}[\|\boldsymbol{F}\boldsymbol{z}_{k+1/2} - \boldsymbol{F}(\boldsymbol{z}_k + \beta_k(\boldsymbol{z}_0 - \boldsymbol{z}_k) - (1 - \beta_k)\alpha_k\boldsymbol{F}\boldsymbol{z}_k)\| \, \|\tilde{\boldsymbol{F}}\boldsymbol{z}_k - \boldsymbol{F}\boldsymbol{z}_k\|] \\
&\leq \mathbb{E}[L\|\boldsymbol{z}_{k+1/2} - (\boldsymbol{z}_k + \beta_k(\boldsymbol{z}_0 - \boldsymbol{z}_k) - (1 - \beta_k)\alpha_k\boldsymbol{F}\boldsymbol{z}_k)\| \, \|\tilde{\boldsymbol{F}}\boldsymbol{z}_k - \boldsymbol{F}\boldsymbol{z}_k\|] \\
&= \mathbb{E}[L(1 - \beta_k)\alpha_k\|\tilde{\boldsymbol{F}}\boldsymbol{z}_k - \boldsymbol{F}\boldsymbol{z}_k\|^2] \\
&= L(1 - \beta_k)\alpha_k\sigma_k^2,
\end{aligned}$$

and

$$|\mathbb{E}[\langle \boldsymbol{F}\boldsymbol{z}_{k+1}, \; \tilde{\boldsymbol{F}}\boldsymbol{z}_{k+1/2} - \boldsymbol{F}\boldsymbol{z}_{k+1/2}\rangle]|$$

$$= |\mathbb{E}[\langle \boldsymbol{F}\boldsymbol{z}_{k+1} - \boldsymbol{F}(\boldsymbol{z}_k + \beta_k(\boldsymbol{z}_0 - \boldsymbol{z}_k) - \alpha_k \boldsymbol{F}\boldsymbol{z}_{k+1/2}), \; \tilde{\boldsymbol{F}}\boldsymbol{z}_{k+1/2} - \boldsymbol{F}\boldsymbol{z}_{k+1/2}\rangle]|$$

$$\leq \mathbb{E}[\|\boldsymbol{F}\boldsymbol{z}_{k+1} - \boldsymbol{F}(\boldsymbol{z}_k + \beta_k(\boldsymbol{z}_0 - \boldsymbol{z}_k) - \alpha_k \boldsymbol{F}\boldsymbol{z}_{k+1/2})\| \, \|\tilde{\boldsymbol{F}}\boldsymbol{z}_{k+1/2} - \boldsymbol{F}\boldsymbol{z}_{k+1/2}\|]$$

$$\leq \mathbb{E}[L\|\boldsymbol{z}_{k+1} - (\boldsymbol{z}_k + \beta_k(\boldsymbol{z}_0 - \boldsymbol{z}_k) - \alpha_k \boldsymbol{F}\boldsymbol{z}_{k+1/2})\| \, \|\tilde{\boldsymbol{F}}\boldsymbol{z}_{k+1/2} - \boldsymbol{F}\boldsymbol{z}_{k+1/2}\|]$$

$$= \mathbb{E}[L\alpha_k\|\tilde{\boldsymbol{F}}\boldsymbol{z}_{k+1/2} - \boldsymbol{F}\boldsymbol{z}_{k+1/2}\|^2]$$

$$= L\alpha_k \sigma_{k+1/2}^2.$$

$\square$

*Proof of Lemma D.1.* First, for $k = 0$, note that

$$V_1 = a_1\|\boldsymbol{F}\boldsymbol{z}_1\|^2 - b_1 \langle \boldsymbol{F}\boldsymbol{z}_1, \, \boldsymbol{z}_0 - \boldsymbol{z}_1\rangle$$

$$= a_1\|\boldsymbol{F}\boldsymbol{z}_1\|^2 - \alpha_0 b_1 \langle \boldsymbol{F}\boldsymbol{z}_1, \, \tilde{\boldsymbol{F}}\boldsymbol{z}_0\rangle$$

$$= \frac{b_1(1-\beta_1)\alpha_1}{2\beta_1}\|\boldsymbol{F}\boldsymbol{z}_1\|^2 - \alpha_0 \langle \boldsymbol{F}\boldsymbol{z}_1, \, \tilde{\boldsymbol{F}}\boldsymbol{z}_0\rangle$$

$$\leq \frac{b_1\alpha_0}{2\beta_0}\|\boldsymbol{F}\boldsymbol{z}_1\|^2 - \alpha_0 \langle \boldsymbol{F}\boldsymbol{z}_1, \, \tilde{\boldsymbol{F}}\boldsymbol{z}_0\rangle$$

$$= \frac{\alpha_0}{2}\|\boldsymbol{F}\boldsymbol{z}_1\|^2 - \alpha_0 \langle \boldsymbol{F}\boldsymbol{z}_1, \, \tilde{\boldsymbol{F}}\boldsymbol{z}_0\rangle. \tag{10}$$

By the given condition, we get

$$0 \leq L_0^2\|\boldsymbol{z}_1 - \boldsymbol{z}_0\|^2 - \|\boldsymbol{F}\boldsymbol{z}_1 - \boldsymbol{F}\boldsymbol{z}_0\|^2 = L_0^2\alpha_0^2\|\tilde{\boldsymbol{F}}\boldsymbol{z}_0\|^2 - \|\boldsymbol{F}\boldsymbol{z}_1 - \boldsymbol{F}\boldsymbol{z}_0\|^2. \tag{11}$$

The sum of (10) and (11) with multiplying factor $\frac{\alpha_0}{2}$ yields

$$V_1 \leq \frac{\alpha_0}{2}\|\boldsymbol{F}\boldsymbol{z}_1\|^2 - \alpha_0 \langle \boldsymbol{F}\boldsymbol{z}_1, \, \tilde{\boldsymbol{F}}\boldsymbol{z}_0\rangle + \frac{\alpha_0}{2}(L_0^2\alpha_0^2\|\tilde{\boldsymbol{F}}\boldsymbol{z}_0\|^2 - \|\boldsymbol{F}\boldsymbol{z}_1 - \boldsymbol{F}\boldsymbol{z}_0\|^2)$$

$$= \frac{\alpha_0}{2}(L^2\alpha_0^2\|\tilde{\boldsymbol{F}}\boldsymbol{z}_0\|^2 - \|\boldsymbol{F}\boldsymbol{z}_0\|^2) - \alpha_0 \langle \boldsymbol{F}\boldsymbol{z}_1, \, \tilde{\boldsymbol{F}}\boldsymbol{z}_0 - \boldsymbol{F}\boldsymbol{z}_0\rangle.$$

Hence, we get

$$V_0 - V_1 \geq \frac{\alpha_0(L^2\alpha_0^2 - 1)}{2}\|\boldsymbol{F}\boldsymbol{z}_0\|^2 - \frac{\alpha_0}{2}(L^2\alpha_0^2\|\tilde{\boldsymbol{F}}\boldsymbol{z}_0\|^2 - \|\boldsymbol{F}\boldsymbol{z}_0\|^2) + \alpha_0 \langle \boldsymbol{F}\boldsymbol{z}_1, \, \tilde{\boldsymbol{F}}\boldsymbol{z}_0 - \boldsymbol{F}\boldsymbol{z}_0\rangle$$

$$= \frac{L^2\alpha_0^3}{2}(\|\boldsymbol{F}\boldsymbol{z}_0\|^2 - \|\tilde{\boldsymbol{F}}\boldsymbol{z}_0\|^2) + \alpha_0 \langle \boldsymbol{F}\boldsymbol{z}_1, \, \tilde{\boldsymbol{F}}\boldsymbol{z}_0 - \boldsymbol{F}\boldsymbol{z}_0\rangle.$$

By taking expectation on the both sides,

$$\mathbb{E}[V_0] - \mathbb{E}[V_1] \geq -\frac{L^2\alpha_0^3}{2}\sigma_0^2 + \alpha_0\mathbb{E}[\langle \boldsymbol{F}\boldsymbol{z}_1, \, \tilde{\boldsymbol{F}}\boldsymbol{z}_0 - \boldsymbol{F}\boldsymbol{z}_0\rangle]$$

$$\geq -\Big(\frac{L^2\alpha_0^3}{2} + L\alpha_0^2\Big)\sigma_0^2$$

where the first inequality uses the fact $\mathbb{E}[\|\boldsymbol{F}\boldsymbol{z}_0\|^2 - \|\tilde{\boldsymbol{F}}\boldsymbol{z}_0\|^2] = -\mathbb{E}[\|\boldsymbol{F}\boldsymbol{z}_0 - \tilde{\boldsymbol{F}}\boldsymbol{z}_0\|^2]$, and the last inequality follows from Lemma D.2. Next, for $k \geq 1$, here we note the following relations for later use:

$$\boldsymbol{z}_{k+1} - \boldsymbol{z}_k = \frac{\beta_k}{1 - \beta_k}(\boldsymbol{z}_0 - \boldsymbol{z}_{k+1}) - \frac{\alpha_k}{1 - \beta_k}\tilde{\boldsymbol{F}}\boldsymbol{z}_{k+1/2},$$

$$\boldsymbol{z}_{k+1} - \boldsymbol{z}_k = \beta_k(\boldsymbol{z}_0 - \boldsymbol{z}_k) - \alpha_k\tilde{\boldsymbol{F}}\boldsymbol{z}_{k+1/2}, \text{ and}$$

$$\boldsymbol{z}_{k+1} - \boldsymbol{z}_{k+1/2} = \alpha_k((1 - \beta_k)\tilde{\boldsymbol{F}}\boldsymbol{z}_k - \tilde{\boldsymbol{F}}\boldsymbol{z}_{k+1/2}).$$

Then, by the given condition, we have

$$V_k - V_{k+1} \geq V_k - V_{k+1} - \frac{b_k}{\beta_k} \langle \boldsymbol{F}\boldsymbol{z}_{k+1} - \boldsymbol{F}\boldsymbol{z}_k, \, \boldsymbol{z}_{k+1} - \boldsymbol{z}_k\rangle$$

$$=V_k - V_{k+1} - \frac{b_k}{\beta_k} \langle \boldsymbol{F} \boldsymbol{z}_{k+1},\, \boldsymbol{z}_{k+1} - \boldsymbol{z}_k \rangle + \frac{b_k}{\beta_k} \langle \boldsymbol{F} \boldsymbol{z}_k,\, \boldsymbol{z}_{k+1} - \boldsymbol{z}_k \rangle$$

$$=(a_k \|\boldsymbol{F} \boldsymbol{z}_k\|^2 - b_k \langle \boldsymbol{F} \boldsymbol{z}_k,\, \boldsymbol{z}_0 - \boldsymbol{z}_k \rangle) - (a_{k+1} \|\boldsymbol{F} \boldsymbol{z}_{k+1}\|^2 - b_{k+1} \langle \boldsymbol{F} \boldsymbol{z}_{k+1},\, \boldsymbol{z}_0 - \boldsymbol{z}_{k+1} \rangle)$$

$$- \frac{b_k}{\beta_k} \left\langle \boldsymbol{F} \boldsymbol{z}_{k+1},\, \frac{\beta_k}{1 - \beta_k}(\boldsymbol{z}_0 - \boldsymbol{z}_{k+1}) - \frac{\alpha_k}{1 - \beta_k} \tilde{\boldsymbol{F}} \boldsymbol{z}_{k+1/2} \right\rangle$$

$$+ \frac{b_k}{\beta_k} \langle \boldsymbol{F} \boldsymbol{z}_k,\, \beta_k(\boldsymbol{z}_0 - \boldsymbol{z}_k) - \alpha_k \tilde{\boldsymbol{F}} \boldsymbol{z}_{k+1/2} \rangle$$

$$=a_k \|\boldsymbol{F} \boldsymbol{z}_k\|^2 - a_{k+1} \|\boldsymbol{F} \boldsymbol{z}_{k+1}\|^2$$

$$+ \left( b_{k+1} - \frac{b_k}{1 - \beta_k} \right) \langle \boldsymbol{F} \boldsymbol{z}_{k+1},\, \boldsymbol{z}_0 - \boldsymbol{z}_{k+1} \rangle + \frac{b_k \alpha_k}{\beta_k(1 - \beta_k)} \langle \boldsymbol{F} \boldsymbol{z}_{k+1},\, \tilde{\boldsymbol{F}} \boldsymbol{z}_{k+1/2} \rangle$$

$$- \frac{\alpha_k b_k}{\beta_k} \langle \boldsymbol{F} \boldsymbol{z}_k,\, \tilde{\boldsymbol{F}} \boldsymbol{z}_{k+1/2} \rangle$$

$$=a_k \|\boldsymbol{F} \boldsymbol{z}_k\|^2 - a_{k+1} \|\boldsymbol{F} \boldsymbol{z}_{k+1}\|^2$$

$$+ \frac{b_k \alpha_k}{\beta_k(1 - \beta_k)} \langle \boldsymbol{F} \boldsymbol{z}_{k+1},\, \tilde{\boldsymbol{F}} \boldsymbol{z}_{k+1/2} \rangle - \frac{\alpha_k b_k}{\beta_k} \langle \boldsymbol{F} \boldsymbol{z}_k,\, \tilde{\boldsymbol{F}} \boldsymbol{z}_{k+1/2} \rangle. \qquad \left( \because b_{k+1} = \frac{b_k}{1 - \beta_k}. \right)$$

$$\tag{12}$$

By the given condition, we get

$$0 \geq \|\boldsymbol{F} \boldsymbol{z}_{k+1} - \boldsymbol{F} \boldsymbol{z}_{k+1/2}\|^2 - L_k^2 \|\boldsymbol{z}_{k+1} - \boldsymbol{z}_{k+1/2}\|^2$$

$$= \|\boldsymbol{F} \boldsymbol{z}_{k+1} - \boldsymbol{F} \boldsymbol{z}_{k+1/2}\|^2 - L_k^2 \alpha_k^2 \|(1 - \beta_k) \tilde{\boldsymbol{F}} \boldsymbol{z}_k - \tilde{\boldsymbol{F}} \boldsymbol{z}_{k+1/2}\|^2. \tag{13}$$

Hence, the sum of (12) and (13) with multiplying factor $\frac{b_k}{2 L_k^2 \alpha_k \beta_k (1 - \beta_k)}$ yields

$$V_k - V_{k+1} \geq a_k \|\boldsymbol{F} \boldsymbol{z}_k\|^2 - a_{k+1} \|\boldsymbol{F} \boldsymbol{z}_{k+1}\|^2$$

$$+ \frac{b_k \alpha_k}{\beta_k(1 - \beta_k)} \langle \boldsymbol{F} \boldsymbol{z}_{k+1},\, \tilde{\boldsymbol{F}} \boldsymbol{z}_{k+1/2} \rangle - \frac{\alpha_k b_k}{\beta_k} \langle \boldsymbol{F} \boldsymbol{z}_k,\, \tilde{\boldsymbol{F}} \boldsymbol{z}_{k+1/2} \rangle$$

$$+ \frac{b_k}{2 L^2 \alpha_k \beta_k (1 - \beta_k)} (\|\boldsymbol{F} \boldsymbol{z}_{k+1} - \boldsymbol{F} \boldsymbol{z}_{k+1/2}\|^2 - L_k^2 \alpha_k^2 \|(1 - \beta_k) \tilde{\boldsymbol{F}} \boldsymbol{z}_k - \tilde{\boldsymbol{F}} \boldsymbol{z}_{k+1/2}\|^2)$$

$$= \left( a_k \|\boldsymbol{F} \boldsymbol{z}_k\|^2 - \frac{b_k(1 - \beta_k)\alpha_k}{2\beta_k} \|\tilde{\boldsymbol{F}} \boldsymbol{z}_k\|^2 \right) + \left( \frac{b_k}{2 L^2 \alpha_k \beta_k (1 - \beta_k)} - a_{k+1} \right) \|\boldsymbol{F} \boldsymbol{z}_{k+1}\|^2$$

$$+ \frac{b_k}{2 L^2 \alpha_k \beta_k (1 - \beta_k)} \|\boldsymbol{F} \boldsymbol{z}_{k+1/2}\|^2 - \frac{b_k \alpha_k}{2\beta_k(1 - \beta_k)} \|\tilde{\boldsymbol{F}} \boldsymbol{z}_{k+1/2}\|^2$$

$$- \frac{b_k}{L^2 \alpha_k \beta_k (1 - \beta_k)} \langle \boldsymbol{F} \boldsymbol{z}_{k+1},\, \boldsymbol{F} \boldsymbol{z}_{k+1/2} \rangle + \frac{b_k \alpha_k}{\beta_k(1 - \beta_k)} \langle \boldsymbol{F} \boldsymbol{z}_{k+1},\, \tilde{\boldsymbol{F}} \boldsymbol{z}_{k+1/2} \rangle$$

$$+ \frac{b_k \alpha_k}{\beta_k} \langle \tilde{\boldsymbol{F}} \boldsymbol{z}_k - \boldsymbol{F} \boldsymbol{z}_k,\, \tilde{\boldsymbol{F}} \boldsymbol{z}_{k+1/2} \rangle$$

By the given conditions, we get $a_k = \frac{b_k(1 - \beta_k)\alpha_k}{2\beta_k}$ and

$$a_{k+1} = \frac{b_{k+1}(1 - \beta_{k+1})\alpha_{k+1}}{2\beta_{k+1}} \leq \frac{b_{k+1}\alpha_k}{2\beta_k} = \frac{b_k \alpha_k}{2\beta_k(1 - \beta_k)}. \qquad \left( \because b_{k+1} = \frac{b_k}{1 - \beta_k}. \right)$$

Therefore, we get

$$V_k - V_{k+1} \geq \frac{b_k(1 - \beta_k)\alpha_k}{2\beta_k} (\|\boldsymbol{F} \boldsymbol{z}_k\|^2 - \|\tilde{\boldsymbol{F}} \boldsymbol{z}_k\|^2) + \frac{b_k(1 - L^2 \alpha_k^2)}{2 L^2 \alpha_k \beta_k (1 - \beta_k)} \|\boldsymbol{F} \boldsymbol{z}_{k+1}\|^2$$

$$+ \frac{b_k}{2 L^2 \alpha_k \beta_k (1 - \beta_k)} \|\boldsymbol{F} \boldsymbol{z}_{k+1/2}\|^2 - \frac{b_k \alpha_k}{2\beta_k(1 - \beta_k)} \|\tilde{\boldsymbol{F}} \boldsymbol{z}_{k+1/2}\|^2$$

$$- \frac{b_k}{L^2 \alpha_k \beta_k (1 - \beta_k)} \langle \boldsymbol{F} \boldsymbol{z}_{k+1},\, \boldsymbol{F} \boldsymbol{z}_{k+1/2} \rangle + \frac{b_k \alpha_k}{\beta_k(1 - \beta_k)} \langle \boldsymbol{F} \boldsymbol{z}_{k+1},\, \tilde{\boldsymbol{F}} \boldsymbol{z}_{k+1/2} \rangle$$

$$+ \frac{b_k \alpha_k}{\beta_k} \langle \tilde{\boldsymbol{F}} \boldsymbol{z}_k - \boldsymbol{F} \boldsymbol{z}_k,\, \tilde{\boldsymbol{F}} \boldsymbol{z}_{k+1/2} \rangle$$

$$= \frac{b_k(1-\beta_k)\alpha_k}{2\beta_k}(\|\boldsymbol{F}\boldsymbol{z}_k\|^2 - \|\tilde{\boldsymbol{F}}\boldsymbol{z}_k\|^2) + \frac{b_k(1-L^2\alpha_k^2)}{2L^2\alpha_k\beta_k(1-\beta_k)}\|\boldsymbol{F}\boldsymbol{z}_{k+1}\|^2$$

$$+ \frac{b_k(1-L^2\alpha_k^2)}{2L^2\alpha_k\beta_k(1-\beta_k)}\|\boldsymbol{F}\boldsymbol{z}_{k+1/2}\|^2 + \frac{b_k\alpha_k}{2\beta_k(1-\beta_k)}(\|\boldsymbol{F}\boldsymbol{z}_{k+1/2}\|^2 - \|\tilde{\boldsymbol{F}}\boldsymbol{z}_{k+1/2}\|^2)$$

$$- \frac{b_k(1-L^2\alpha_k^2)}{L^2\alpha_k\beta_k(1-\beta_k)}\langle \boldsymbol{F}\boldsymbol{z}_{k+1}, \boldsymbol{F}\boldsymbol{z}_{k+1/2}\rangle + \frac{b_k\alpha_k}{\beta_k(1-\beta_k)}\langle \boldsymbol{F}\boldsymbol{z}_{k+1}, \tilde{\boldsymbol{F}}\boldsymbol{z}_{k+1/2} - \boldsymbol{F}\boldsymbol{z}_{k+1/2}\rangle$$

$$+ \frac{b_k\alpha_k}{\beta_k}\langle \tilde{\boldsymbol{F}}\boldsymbol{z}_k - \boldsymbol{F}\boldsymbol{z}_k, \tilde{\boldsymbol{F}}\boldsymbol{z}_{k+1/2}\rangle$$

$$= \frac{b_k(1-\beta_k)\alpha_k}{2\beta_k}(\|\boldsymbol{F}\boldsymbol{z}_k\|^2 - \|\tilde{\boldsymbol{F}}\boldsymbol{z}_k\|^2) + \frac{b_k\alpha_k}{2\beta_k(1-\beta_k)}(\|\boldsymbol{F}\boldsymbol{z}_{k+1/2}\|^2 - \|\tilde{\boldsymbol{F}}\boldsymbol{z}_{k+1/2}\|^2)$$

$$+ \frac{b_k(1-L^2\alpha_k^2)}{2L^2\alpha_k\beta_k(1-\beta_k)}\|\boldsymbol{F}\boldsymbol{z}_{k+1} - \boldsymbol{F}\boldsymbol{z}_{k+1/2}\|^2$$

$$+ \frac{b_k\alpha_k}{\beta_k(1-\beta_k)}\langle \boldsymbol{F}\boldsymbol{z}_{k+1}, \tilde{\boldsymbol{F}}\boldsymbol{z}_{k+1/2} - \boldsymbol{F}\boldsymbol{z}_{k+1/2}\rangle + \frac{b_k\alpha_k}{\beta_k}\langle \tilde{\boldsymbol{F}}\boldsymbol{z}_k - \boldsymbol{F}\boldsymbol{z}_k, \tilde{\boldsymbol{F}}\boldsymbol{z}_{k+1/2}\rangle$$

$$\geq \frac{b_k(1-\beta_k)\alpha_k}{2\beta_k}(\|\boldsymbol{F}\boldsymbol{z}_k\|^2 - \|\tilde{\boldsymbol{F}}\boldsymbol{z}_k\|^2) + \frac{b_k\alpha_k}{2\beta_k(1-\beta_k)}(\|\boldsymbol{F}\boldsymbol{z}_{k+1/2}\|^2 - \|\tilde{\boldsymbol{F}}\boldsymbol{z}_{k+1/2}\|^2)$$

$$+ \frac{b_k\alpha_k}{\beta_k(1-\beta_k)}\langle \boldsymbol{F}\boldsymbol{z}_{k+1}, \tilde{\boldsymbol{F}}\boldsymbol{z}_{k+1/2} - \boldsymbol{F}\boldsymbol{z}_{k+1/2}\rangle + \frac{b_k\alpha_k}{\beta_k}\langle \tilde{\boldsymbol{F}}\boldsymbol{z}_k - \boldsymbol{F}\boldsymbol{z}_k, \tilde{\boldsymbol{F}}\boldsymbol{z}_{k+1/2}\rangle.$$

Then by taking expectation on the both sides and using the fact $\mathbb{E}[\langle \tilde{\boldsymbol{F}}\boldsymbol{z}_k - \boldsymbol{F}\boldsymbol{z}_k, \tilde{\boldsymbol{F}}\boldsymbol{z}_{k+1/2}\rangle] = \mathbb{E}[\langle \tilde{\boldsymbol{F}}\boldsymbol{z}_k - \boldsymbol{F}\boldsymbol{z}_k, \boldsymbol{F}\boldsymbol{z}_{k+1/2}\rangle]$, we get

$$\mathbb{E}[V_k] - \mathbb{E}[V_{k+1}] \geq -\frac{b_k(1-\beta_k)\alpha_k}{2\beta_k}\sigma_k^2 - \frac{b_k\alpha_k}{2\beta_k(1-\beta_k)}\sigma_{k+1/2}^2$$

$$+ \frac{b_k\alpha_k}{\beta_k(1-\beta_k)}\mathbb{E}[\langle \boldsymbol{F}\boldsymbol{z}_{k+1}, \tilde{\boldsymbol{F}}\boldsymbol{z}_{k+1/2} - \boldsymbol{F}\boldsymbol{z}_{k+1/2}\rangle] + \frac{b_k\alpha_k}{\beta_k}\mathbb{E}[\langle \tilde{\boldsymbol{F}}\boldsymbol{z}_k - \boldsymbol{F}\boldsymbol{z}_k, \boldsymbol{F}\boldsymbol{z}_{k+1/2}\rangle]$$

$$\geq -\frac{b_k\alpha_k(1+2L\alpha_k)}{2\beta_k}\left((1-\beta_k)\sigma_k^2 + \frac{1}{1-\beta_k}\sigma_{k+1/2}^2\right)$$

where the last inequality follows from Lemma D.2. $\qquad\square$

## D.3 Proof of Theorem 6.1

Note that S-FEG is equivalent to (Class S-FEG) with $\alpha_k = \frac{1}{L}$ and $\beta_k = \frac{1}{k+1}$. It is straightforward to verify that the given $\{\alpha_k\}_{k\geq 0}$ and $\{\beta_k\}_{k\geq 0}$ satisfy the conditions in Lemma D.1 for all $k \geq 0$. By noting that

$$a_k = \frac{b_k(1-\beta_k)\alpha_k}{2\beta_k} = \frac{k^2}{2L} \qquad \text{and}$$

$$b_k = \frac{1}{1-\beta_{k-1}}b_{k-1} = \Big(\prod_{i=1}^{k-1}\frac{1}{1-\beta_i}\Big)b_1 = k,$$

Lemma D.1 implies that

$$\mathbb{E}[V_k] = \mathbb{E}[V_k - V_0]$$

$$= \sum_{l=0}^{k-1}(\mathbb{E}[V_{l+1}] - \mathbb{E}[V_l])$$

$$\leq \Big(\frac{L^2\alpha_0^3}{2} + L\alpha_0^2\Big)\sigma_0^2 + \sum_{l=1}^{k-1}\frac{b_l\alpha_l(1+2L\alpha_l)}{2\beta_l}\Big((1-\beta_l)\sigma_l^2 + \frac{1}{1-\beta_l}\sigma_{l+1/2}^2\Big)$$

$$= \frac{3}{2L}\sigma_0^2 + \sum_{l=1}^{k-1}\frac{3}{2L}(l^2\sigma_l^2 + (l+1)^2\sigma_{l+1/2}^2)$$

$$\overset{\text{let}}{=} \sigma_{\text{total}}^2.$$

Therefore, noting that $\mathbb{E}[V_k] = \mathbb{E}\left[\frac{k^2}{2L}\|\boldsymbol{F}\boldsymbol{z}_k\|^2 - k\langle\boldsymbol{F}\boldsymbol{z}_k,\,\boldsymbol{z}_0 - \boldsymbol{z}_k\rangle\right]$, we get

$$\mathbb{E}\left[\frac{k^2}{2L}\|\boldsymbol{F}\boldsymbol{z}_k\|^2\right] \le \mathbb{E}[k\langle\boldsymbol{F}\boldsymbol{z}_k,\,\boldsymbol{z}_0 - \boldsymbol{z}_k\rangle] + \sigma_{\text{total}}^2$$
$$= \mathbb{E}[k\langle\boldsymbol{F}\boldsymbol{z}_k,\,\boldsymbol{z}_0 - \boldsymbol{z}_*\rangle + k\langle\boldsymbol{F}\boldsymbol{z}_k,\,\boldsymbol{z}_* - \boldsymbol{z}_k\rangle] + \sigma_{\text{total}}^2$$
$$= \mathbb{E}[k\langle\boldsymbol{F}\boldsymbol{z}_k,\,\boldsymbol{z}_0 - \boldsymbol{z}_*\rangle] + \sigma_{\text{total}}^2 \qquad (\because \text{monotonicity of } \boldsymbol{F})$$
$$\le \mathbb{E}\left[\frac{k^2}{4L}\|\boldsymbol{F}\boldsymbol{z}_k\|^2 + L\|\boldsymbol{z}_0 - \boldsymbol{z}_*\|^2\right] + \sigma_{\text{total}}^2. \qquad (\because \text{Young's inequality.})$$

As a result, we get $\mathbb{E}\left[\frac{k^2}{4L}\|\boldsymbol{F}\boldsymbol{z}_k\|^2\right] \le L\|\boldsymbol{z}_0 - \boldsymbol{z}_*\|^2 + \frac{3}{2L}\left[\sigma_0^2 + \sum_{l=1}^{k-1}(l^2\sigma_l^2 + (l+1)^2\sigma_{l+1/2}^2)\right]$ and, by dividing the both sides by $\frac{k^2}{4L}$, we get

$$\mathbb{E}[\|\boldsymbol{F}\boldsymbol{z}_k\|^2] \le \frac{4L^2\|\boldsymbol{z}_0 - \boldsymbol{z}_*\|^2}{k^2} + \frac{6}{k^2}\left[\sigma_0^2 + \sum_{l=1}^{k-1}(l^2\sigma_l^2 + (l+1)^2\sigma_{l+1/2}^2)\right].$$

In addition, if $\sigma_0^2 \le \frac{\epsilon}{6}$, $\sigma_k^2 \le \frac{\epsilon}{6k}$ and $\sigma_{k+1/2}^2 \le \frac{\epsilon}{6(k+1)}$ for all $k \ge 1$, then we have

$$\mathbb{E}[\|\boldsymbol{F}\boldsymbol{z}_k\|^2] \le \frac{4L^2\|\boldsymbol{z}_0 - \boldsymbol{z}_*\|^2}{k^2} + \frac{6}{k^2}\left[\frac{\epsilon}{6} + \sum_{l=1}^{k-1}\left(\frac{\epsilon l}{6} + \frac{\epsilon(l+1)}{6}\right)\right]$$
$$= \frac{4L^2\|\boldsymbol{z}_0 - \boldsymbol{z}_*\|^2}{k^2} + \frac{\epsilon}{k^2}\sum_{l=0}^{k-1}(2l+1)$$
$$= \frac{4L^2\|\boldsymbol{z}_0 - \boldsymbol{z}_*\|^2}{k^2} + \frac{\epsilon}{k^2}\sum_{l=0}^{k-1}((l+1)^2 - l^2)$$
$$= \frac{4L^2\|\boldsymbol{z}_0 - \boldsymbol{z}_*\|^2}{k^2} + \epsilon.$$

$\square$