# OpenReview forum: "Fast Extra Gradient Methods for Smooth Structured Nonconvex-Nonconcave Minimax Problems"
_NeurIPS.cc/2021/Conference — NeurIPS 2021 Poster_

### Official Review · Reviewer_VnM6 · 2021-07-15

**Rating:** 8
**Confidence:** 4

**Summary:**

This paper proposes and analyses a variant of Extragradient with anchoring and reuse of past gradients that they call Class FEG. They analyze this method in the context of Lipschitz and comonotone operators (that is a subclass of non-monotone operators). They also propose a version of their algorithm with backtracking line-search and propose an analysis in the stochastic setting (with vanishing noise!). They show accelerated (with respect to the normal EG rates) convergence rates in the comonotone setting when the comonotonicity is small enough with respect to the Lipschitness of the operator.

**Limitations And Societal Impact:**

Yes

**Main Review:**

### Main Points:
- Significance: This paper proposes a new method with accelerated rates in a non-monotone setting. It is very relevant to the ML community since it deals with the comonotone setting, which is a first step to understand the general non-monotone setting that has application to GANs and, more generally, any game with non-convex payoffs.
Quality: This paper provides high-quality results. I have a question about Lemma 8.1 that I would like the authors to address: First of all, I would like to know how the authors address the previous works that proposed worst-case scenarios for smooth operators ([Hsieh et al. 2021] and [Letcher et al. 2021]). Second of all, I am a bit confused because Lemma 8.1 seems to provide a counterexample for Theorem 4.1. Since what only matters are what happens in the line {x=y} and that on that line, F is constant (hence Lipschitz and comonotone). Even if the example provided in Lemma 8.1 is not comonotone (which is currently not proven in the paper), it is not clear to me why it is impossible to extend a vector field F such that $F(x,x) = (-\sqrt{LR},-\sqrt{LR}), \forall x$ into a Lipschitz and comonotone vector field.

- Clarity: The paper is very clear and well written
- Originality: The proposed method is novel, and the analysis with accelerated rates is new and very relevant to the community.
The main weakness of the paper is that there is no experiment to illustrate the performance of their algorithm in such a \rho-comonote case. The consequence of this lack of experiment is two-fold: first, the authors do not propose any concrete example of realistic or synthetic comonontone games for which their assumptions hold (namely $\rho > -\frac1{2L}$). Secondly, it would illustrate the practical improvement over Extragrdient in such a setting and the additive value of the proposed backtracking line search.

### Conclusion:
Besides the weakness mentioned above, I think this paper is a strong contribution to the community and should be accepted to NeurIPS. However, if the authors could provide some experimental results and answer my question about Lemma 7.1, I would consider increasing my grade.

### Minor comments:
L239: $V_1$ is not equal to 0! I checked your proof, and thanks to the fact that $\alpha \leq 1/L$ we have $V_0 \leq 0$.

### Reference:
Hsieh, Ya-Ping, Panayotis Mertikopoulos, and Volkan Cevher. "The limits of min-max optimization algorithms: Convergence to spurious non-critical sets." International Conference on Machine Learning. PMLR, 2021.
Letcher, Alistair. "On the impossibility of global convergence in multi-loss optimization." ICLR 2021.

### After Rebuttal
After the discussion with the authors, I decided to maintain my score.


**Time Spent Reviewing:**

4

---

> ### Author Response · Authors · 2021-08-10
> **Reply to Reviewer VnM6**
>
> We appreciate the reviewer for understanding the importance of our problem settings and the significance of our contributions.
>
> $\bullet$ **(Worst-case scenarios for smooth operators):**
> [Hsieh et al. 2021] and [Letcher et al. 2021]
> described the fundamental limits for solving the smooth minimax problems under the weak asymptotic coercivity and the coercivity, respectively. Both studies focused on the worst-case that iteration points cycle around a stationary point without converging to it.
> It is worth noting that
> our smooth minimax example in Lemma 8.1 does not satisfy neither
> the weak asymptotic coercivity nor the coercivity,
> but satisfies the function boundedness,
> i.e., $\max_{x,y} f(x,y) - \min_{x,y} f(x,y) \le R$.
> Another interesting aspect of our example is
> that all points in $\big\\{(x,y)\big|x\le y- \sqrt{\frac{R}{L}}$ or $x\ge y+ \sqrt{\frac{R}{L}}\big\\}$
> are stationary points.
>
> $\bullet$ **(Discussion on Lemma 8.1):**
> This is a very interesting point!
> We were able to show that
> an example in Lemma 8.1 is not comonotone,
> and more importantly,
> that there does not exist a Lipschitz and comonotone vector field $F$
> that satisfies $F(x,x)=(-\sqrt{LR},-\sqrt{LR})$ for all $x$ and has a stationary point.
> First,
> let $z=\big(x,x+\sqrt{\frac{R}{L}}\big)$ and $w=(0,0)$.
> Since $Fz = (0,0)$ and $Fw = (-\sqrt{LR},-\sqrt{LR})$,
> we get $\langle Fz-Fw,z-w\rangle = 2\sqrt{LR}x+R$ and $||Fz-Fw||^2=2LR$, which implies that $\rho = -\infty$
> in the comonotonicity condition
> as we take $x\rightarrow -\infty$.
> So an example in Lemma 8.1 is not comonotone,
> and thus not a counterexample of Theorem 4.1.
> Second,
> let $z=(s,0)$, for some $s>0$, be a stationary point without loss of generality,
> and let
> $w=(x,x)$.
> Since $Fz = (0,0)$ and $Fw = (-\sqrt{LR},-\sqrt{LR})$,
> we get $\langle Fz-Fw, z-w \rangle  = \sqrt{LR}(s-2x)$ and $||Fz-Fw||^2=2LR$, which implies that $\rho = -\infty$
> in the comonotonicity condition
> as we take $x\rightarrow \infty$.
>
> $\bullet$ **(Experiment):**
> We agree that an experiment can strengthen the paper.
> We ran a toy experiment with a function $f(x,y) = \frac{\rho L^2}{2} x^2 + L\sqrt{1-\rho^2L^2}xy - \frac{\rho L^2}{2} y^2$, which has an $L$-Lipschitz $\rho$-comonotone saddle gradient.
> For $\rho = -1/3L$ and $L=1$, we observed that FEG converges with an accelerated rate, whereas
> EG+, EAG-C, and EAG-V diverge.
> We will add this result in the revision,
> possibly with the line search result.
>
> $\bullet$ **(Minor comments):**
> You are right.
> We appreciate for the detailed comment
> and we will fix it in our revision.

---

> > ### Comment · Reviewer_VnM6 · 2021-08-25
> > **Details on the worst case**
> >
> > Hello,
> >
> > Thank you for your answers.
> > I am a bit confused with your answer regarding the relationship to [Hsieh et al. 2021] and [Letcher et al. 2021]. It seems like your last iterate convergence guarantees may contradict the limit cycles results provided in these two papers.
> >
> > One way to resolve the contradiction would be to show that the negative results from Hsieh and Letcher exclude \rho-comonotone vector fields. Is it actually the case?
> > Or is there another reason that justify that your results and the ones from [Hsieh et al. 2021] and [Letcher 2021] are not contradicting each other.
> >
> > Best, Reviewer VnM6

---

> > > ### Author Response · Authors · 2021-08-27
> > > **Contradiction?**
> > >
> > > We now see what exactly you were curious about.
> > > First of all, if there is a contradiction between
> > > the convergence guarantee of FEG (under the negative comonotonicty)
> > > and the claims in [Hsieh et al. 2021, Letcher et al. 2021],
> > > then similarly the convergence guarantee of EG+ in [7]
> > > (under the weak MVI that is weaker than the negative comonotonicity)
> > > will also be in trouble,
> > > which is not the case.
> > > To make a long story short,
> > > the classes of methods considered
> > > in [Hsieh et al. 2021, Letcher et al. 2021]
> > > do not include both EG+ and FEG.
> > > In addition, [Hsieh et al. 2021, Letcher et al. 2021]
> > > exhibit the limit of training methods
> > > by introducing "worst-case" functions
> > > that the considered algorithms do not find stationary points.
> > > We noticed that such "worst-case" functions are not smooth
> > > (i.e., their gradients are not Lipschitz continuous),
> > > so we can again see that
> > > the claims in [Hsieh et al. 2021, Letcher et al. 2021]
> > > do not contradict with the guarantees of EG+ and FEG.
> > > (We would like to mention that
> > > we should have dropped the term "smooth"
> > > in our previous response
> > > that states that [Hsieh et al. 2021, Letcher et al. 2021]
> > > study the limits for solving the "smooth" minimax problems.)

---

> > > > ### Comment · Area_Chair_QjAG · 2021-08-27
> > > > **line-search**
> > > >
> > > > Dear Authors,
> > > >
> > > > Can you also comment whether or not you can prove your linesearch step will succeed in finite number of steps?
> > > >
> > > > best,
> > > > AC

---

> > > > > ### Author Response · Authors · 2021-08-27
> > > > > **line-search works**
> > > > >
> > > > > Dear AC,
> > > > >
> > > > > You can find the proof in Lemma 5.1 (and lines 188-189). This basically says that the line search step will terminate in finite number of steps, due to the positive lower boundedness of $\tau_k$ and $\eta_k$ in FEG-A under the condition $\rho>-\frac{1-\delta}{2L}$. This was not super clear and we will clarify this in the revision.
> > > > >
> > > > > FYI, while answering your question, we noticed that the last inequality in Algorithm 2 is in opposite direction. We will fix it in our revision.
> > > > >
> > > > > Best,
> > > > >
> > > > > Authors

---

> > > > > > ### Comment · Area_Chair_QjAG · 2021-08-27
> > > > > > **thanks for the clarification and the pointer.**
> > > > > >
> > > > > > This is helpful.
> > > > > >
> > > > > > best,
> > > > > > AC

---

### Official Review · Reviewer_ifPf · 2021-07-16

**Rating:** 6
**Confidence:** 4

**Summary:**

This paper proposes a two-time-scale and anchored extragradient method for smooth structured nonconvex-nonconcave problems. The proposed FEG method with the Lipschitz continuous and negative comonotone operators has an accelerated convergence rate, O(1/k^2), with respect to the squared gradient norm.. Moreover, the authors further study the backtracking line-search version of FEG, named FEG-A, for smooth structured nonconvex-nonconcave problems, and also propose its stochastic version, named S-FEG, for smooth convex-concave problems.

**Limitations And Societal Impact:**

1.	In Theorem 4.1, the condition \rho>-1/2L needs to be modified, because when \rho=1/sqrt(k)-1/2L, the convergence rate is O(1/k) instead of O(1/k^2).
2.	The update rules in Algorithm 1 depends on the parameter \rho, for a practical min-max problem. How to determine the parameter \rho?
3.	Although the paper is theoretically sound, there is a lack of experiments to verify the proposed algorithms.

**Main Review:**

The paper is nicely written and overall easy to follow. The rigorous convergence analysis of the proposed algorithms has been provided in this paper.

There are several major issues in this paper:
1.	In Theorem 4.1, the condition \rho>-1/2L needs to be modified, because when \rho=1/sqrt(k)-1/2L, the convergence rate is O(1/k) instead of O(1/k^2).
2.	The update rules in Algorithm 1 depends on the parameter \rho.  In a practical min-max problem, how to determine the parameter \rho?
3.	Although the paper is theoretically sound, there is a lack of experiments to verify the proposed algorithms.


**Time Spent Reviewing:**

72

---

> ### Author Response · Authors · 2021-08-10
> **Reply to Reviewer ifPf**
>
> We would like to thank the reviewer for
> the nice summary of our work,
> and below is our response to the reviewer's concern.
>
> 1. The comonotonicity parameter, $\rho$, is a fixed given constant,
> so it should not include $k$ that changes over iterations.
> If we let $\rho = \frac{1}{\sqrt{n}}-\frac{1}{2L}$ for a fixed positive value $n$, FEG satisfies $||Fz_k||^2 \le n||z_0-z_*||^2/k^2 = O(1/k^2)$ for all $k$ by Theorem 4.1.
> 2. When the global parameters, $L$ and $\rho$, are unknown, we update the parameters of FEG adaptively using the backtracking line-search technique, as illustrated in Section 5.
> 3. We appreciate the reviewer for considering our paper
> to be theoretically sound.
> However, we do not agree that a lack of experiments
> is a major issue
> and a justification for a low score
> on a theory-oriented paper.
> As requested, we will add a nonconvex-nonconcave example,
> which the FEG converges with rate $O(1/k^2)$, but EG+, EAG-C, and EAG-V diverge.

---

> ### Comment · Reviewer_ifPf · 2021-09-02
> **Change of score**
>
> I increased my score, but I still think the assumption 2 is the main reason to obtain $O(1/k^2)$ convergence rate. Does the function satisfying the assumption really exist for a real nonconvex nonconcave min-max problem? If not, what is the significance of the theory? When n is large, although the convergence rate is  $O(1 / k^2)$, the complexity is still high.

---

> > ### Author Response · Authors · 2021-09-03
> > **Reply to Reviewer ifPf**
> >
> > We thank the reviewer for the response and raising the score.
> >
> > - **(assumptions to guarantee $O(1/k^2)$):**
> > [38] used assumption 2 with $\rho=0$ (convex-concave case) to achieve $O(1/k^2)$,
> > which is shown to be optimal in [38].
> > Our main contribution is to preserve such fast $O(1/k^2)$ rate
> > under a "weaker" (nonconvex-nonconcave) condition that allows negative $\rho$.
> > We think that the reviewer is having an opposite view.
> > We also understand that one might view that
> > adding assumption 2, in addition to assumption 1,
> > is some artificial to guarantee fast rate.
> > We expected such response, and that is why we added section 8
> > to relieve such concern
> > by showing that without conditions like assumption 2
> > one will not even find a stationary point.
> > However, we think that the theory under nonconvex-nonconcave setting (including our work)
> > is yet not mature enough,
> > and we believe further investigation will fully resolve your concern,
> > which we promise to do in near future.
> >
> > - **(Practicality of the assumption):** We admit that our work and other related works
> > for structured nonconvex-nonconcave problems
> > have limited applications yet,
> > although they are theoretically interesting
> > in terms of expanding our perspective on nonconvex-nonconcave problems.
> > We promise to fill this gap in near future.
> >
> > - **(high complexity?):**
> > We believe that your concern will only be resolved
> > by weakening the condition $\rho > - \frac{1}{2L}$.
> > We agree that such restriction is a limitation,
> > but we would like to again emphasize
> > that there is no other known explicit method
> > that works for the case $\rho = \frac{1}{\sqrt{n}} - \frac{1}{2L}$
> > with large $n$.

---

### Official Review · Reviewer_NehK · 2021-07-18

**Rating:** 6
**Confidence:** 4

**Summary:**

This paper develops a new extra gradient algorithm for solving minimax problems with negative comonotonicity condition. This condition allows nonmonotonicity and therefore covers a class of nonconvex nonconcave minimax problems. This method gets 1/k^2 rate on the squared gradient norm. The algorithm builds on the recent anchored extra gradient algorithm by Yoon, Ryu [39] which focused on monotone case. For handling nonmonotone case, the authors use the recent development by Diakonilas et al [7] that focused on weakly Minty VI (MVI) problems which is a more general template than negative comonotone. This paper also includes extensions with line search and stochastic oracles. For the monotone case, the bounds in this paper are tighter than [39] and the analysis is simpler.

**Limitations And Societal Impact:**

The work does not have potential negative societal impacts. In the other hand, I suggest the authors to highlights the limits in the paper in terms of existence/lack of problems satisfying negative comonotonicity, or flexibility of the analysis: why does MVI and weak MVI assumptions not sufficient?

**Main Review:**

In terms of techniques, this paper mostly builds on [39]. When \rho = 0, FEG is slightly different from EAG of [39]. The change is that the step size for the first update is (1-\beta_k)\alpha_k instead of \alpha_k. To handle negative comonotonicity, FEG incorporates additional terms depending on \rho_k and Fz_k. This modification is used in the analysis to remove the additional error terms coming from lack of monotonicity (as in line 447, first line: -\rho_k\|Fz_{k+1} - Fz_k \|^2.

From what I understand, the authors cannot handle weak MVI assumption since the proof of [39] uses monotonicity with z_{k+1} and z_k, which corresponds here to the first line in line 447. Therefore, this drawback is inherited from the analysis of [39]. Is this correct? On this front, I am concerned with the novelty because of this. The analysis seems to be slightly tighter and giving better bound, but there is no advantage on flexibility: the paper cannot handle MVI or weak MVI assumptions.

The extensions for line search and stochastic oracle seem straightforward but these are not claimed to be the main contributions, so it is fine. More interesting for me is that the authors managed to derive a tight rate compared to [39]. On the other hand, I am curious how this translates to practice. Can the authors run a toy example and compare with EAG-V?

Presentation: I think the paper highlights nonconvex-nonconcave problems too much that it can be misleading. The paper can only handle a small class of nonconvex-nonconcave problems, whereas the abstract and the intro are written too much highlighting this aspect, using GANs, adversarial learning etc. However in the sequel, the authors do not mention at all if the negative comononicity holds for any of the problems. Moreover, the paper cannot handle weak MVI or MVI problems which are themselves restrictive in terms of applications.

Examples: The authors only use Example 1 to give example problems for Assumption 2. Is there any interesting concrete examples that satisfy this assumption? Otherwise, I recommend authors to decrease the focus of introduction on nonconvexity. If there are no interesting applications of the assumption, I doubt it has much value.

I might increase my score if the authors can highlight novelties in their analysis on top of [39] and clarify whether any concrete applications exist for negative comonotonicity.

**Time Spent Reviewing:**

4

---

> ### Author Response · Authors · 2021-08-10
> **Reply to Reviewer NehK**
>
> We would like to thank the reviewer for
> the nice summary of our work
> and constructive feedback.
>
> $\bullet$ **(MVI and weak MVI):**
> We do not agree that the fact that
> the algorithm cannot handle
> the MVI and the weak MVI conditions
> is a disadvantage (in terms of the flexibility).
> Similarly, no one underrates Nesterov's accelerated method
> for strictly assuming convexity and smoothness of the function,
> rather than the quasi-convexity (analogous to the MVI).
> It seems that such limitation is inevitable
> for acceleration-based methods.
>
> $\bullet$ **(Toy example):**
> We ran a toy experiment with a function}
> $f(x,y) = \frac{\rho L^2}{2} x^2 + L\sqrt{1-\rho^2L^2}xy - \frac{\rho L^2}{2} y^2$, which has an $L$-Lipschitz $\rho$-comonotone saddle gradient.
> For the case $\rho = -1/3L$ and $L=1$,
> we observed that FEG converges with an accelerated rate,
> whereas
> EG+, EAG-C, and EAG-V diverge.
>
> $\bullet$ **(Highlighting the structured nonconvexity-nonconcavity too much?):**
> We believe that studying the comonotonicity setting is  one of the important steps
> for understanding the training dynamics of
> the modern nonconvex-nonconcave minimax problems
> such as GAN and adversarial training.
>     However, we agree that this was not well explained in the paper,
>     and we feel that explaining
>     the $\alpha\ge0$-interaction dominant condition in [10]
>     yielding the comonotonicity
>     will be useful here.
>     A convex-concave function is $\alpha\ge0$-interaction dominant.
>     On the other hand, for any $\gamma$-weakly-convex-weakly-concave function,
>     the condition holds with $\alpha = -\gamma < 0$.
>     Its extreme case is $\phi(x,y) = -\frac{\gamma}{2}x^2 + \frac{\gamma}{2}y^2$,
>     where there is no interaction between $x$ and $y$.
>     Interestingly, a nonconvex-nonconcave function
>     becomes $\alpha\ge0$-interaction dominant,
>     when the second term in the condition is sufficiently positive definite.
>     In other words, the condition is satisfied
>     when the interaction term of the Hessian
>     $\nabla^2_{xy}\phi$ is dominating any negative curvature
>     in Hessians $\nabla_{xx}^2\phi$ and $-\nabla_{yy}^2\phi$.
>     In short, the $\alpha\ge0$-interaction dominant condition captures whether
>     there is a sufficient coupling between $x$ and $y$,
>     and we believe many practical (adversarial) applications
>     (at least locally or asymptotically) satisfy such general interaction condition.
> Nevertheless, we will revise our abstract and introduction,
> so that the readers know the specific class of problems we are dealing with
> from the beginning.
>
> $\bullet$ **(Novelty of our work over [38]):**
> While the reviewer considers our line search and stochastic extensions
> to be straightforward results,
> we believe that they are not so trivial as guessed.
> Our analysis is much simpler than that of [38]
> due to the specific formulation of FEG,
> and we want to emphasize that
> this made us to relatively easily study various extensions such as FEG-A and S-FEG.
> In addition, if you are convinced
> with our previous responses on the comonotonicity,
> you will now see that our work is especially novel
> under the specific class of nonconvex-nonconcave problems of interest.

---

> > ### Comment · Reviewer_NehK · 2021-09-02
> > **Change of score**
> >
> > I thank the authors for the response. I read the other discussions as well and as a result am increasing my score to accept.
> >
> > My main criticism is still that the problem class is not motivated very well in terms of applications and the analytic developments (in my personal opinion) are not adding much to what we know from previous papers by [7, 38]. Yet, in my view, the paper still has a concrete contribution on top of [7, 38] which could warrant acceptance.

---

> > > ### Author Response · Authors · 2021-09-03
> > > **Reply to Reviewer NehK**
> > >
> > > We appreciate the reviewer for noticing that we have concrete contribution, and raising the score.
> > > We admit that our work and other related works
> > > for structured nonconvex-nonconcave problems
> > > have limited applications yet,
> > > although they are theoretically interesting.
> > > We promise to fill this gap in near future.

---

### Official Review · Reviewer_KwBn · 2021-07-18

**Rating:** 7
**Confidence:** 4

**Summary:**

Consider unconstrained, possible non-convex-non-concave min-max problems where the saddle gradient operator is Lipschitz and comonotone. Suppose our aim is to minimize the squared norm of the saddle gradient. Inspired by the Halpern iteration and generalized extragradient method, this paper proposes a method called Fast Extragradient (FEG). This paper shows that both FEG and a backtracking variant of it converge at a $O(1/t^2)$ rate, whereas in literature, this rate is only achieved assuming the problem is convex-concave. This paper also proposes a stochastic method called Stochastic FEG (S-FEG); if the variances of the noisy gradients is $O(1/t)$, then S-FEG also converges at a $O(1/k^2)$ rate in expectation.

**Limitations And Societal Impact:**

Please address the issues I raised in the main review.

**Main Review:**

Overall, I think the paper is solid and novel enough. I like the examples and the discussion in Section 8; they are non-trivial and informative. Regarding novelty, the proposed FEG method does not look like a direct combination of the Halpern iteration and extragradient method. Indeed, it is curious to me how the authors found the form of FEG, though I feel it should be possible given, e.g., [6,7,38]. It would be good if the authors can elaborate on how they came up with the idea.

[38], which proposes an FEG-like method for smooth convex-concave problems, is perhaps the most similar work to the submission. Since the convergence guarantees in this submission require a lower bound on the comonotonicity parameter in terms of the smoothness parameter, I would like the authors clarify if the algorithm and analyses in the submission can be directly obtained given [38] or not and, if possible, address the possibility of getting rid of the lower bound on the comonotonicity parameter.

The significance of this work seems to be theoretical. I wonder if there is any practical problem that is not convex-concave and satisfies the comonotonicity condition in the paper.

The S-FEG part seems isolated. The algorithm is not exactly an instance of FEG. It would be good if the authors can explain why. It seems that the authors want to pursue a $O(1/t^2)$ convergence rate. As the stochastic setup includes several learning and statistical problems, I feel it more reasonable to argue that it is in general impossible to have a rate faster than $1/t$.

I do not get the second last line in the equation following Line 239. Please explain.

The presentation is clear. I suggest the authors indicate where the reader can find the proof of Theorem 4.1 after presenting the theorem.

I do not check all proofs in the supplementary material.

Typos:
- Line 104: The lower bound should not be expressed in big-O.
- Line 228: *the* potential function


**Time Spent Reviewing:**

6

---

> ### Author Response · Authors · 2021-08-10
> **Reply to Reviewer KwBn**
>
> We would like to thank the reviewer for
> the nice summary of our work
> and constructive feedback.
>
> $\bullet$ **(How we found FEG?):**
> Although we claim that the class FEG
> are built upon the EG+ and EAG
> (and the specific potential function of EAG),
> we agree that it does not fully explain how
> we found the specific formulation of FEG,
> especially the non-trivial coefficients and the reuse of $Fz_k$.
> Some people regard Nesterov's accelerated method to have
> no clear interpretation and to rely on a "algebraic trick"
> (while there are many recent efforts to understand the method).
> Similarly, our specific formulation of FEG
> mostly relies on a "algebraic trick" yet,
> and we hope to further understand in near future.
>
> $\bullet$ **(Lower bound on the comonotonicity):**
> We understand that one might view
> the lower bound on the comonotonicity as a limitation,
> since such condition is not widely studied yet.
> First of all, EG+ has a tighter lower bound $\rho>-\frac{1}{8L}$,
> unlike our  $\rho>-\frac{1}{2L}$.
> In addition,
> the case $\rho\le -\frac{1}{L}$ is out of
> anyone's interest,
> because it contains
> extreme cases
> such as a "concave-convex" function
> $f(x,y) = -\frac{L}{2}x^2 + \frac{L}{2}y^2$
> without a coupling term.
> We hope you now see that the lower bound is necessary.
> Then, we would like to emphasize that
> deriving a method that works under the negative comonotonicity
> is not a trivial task, even given [38].
> Especially due to our novel formulation of FEG,
> our analysis is much simpler than that of [38],
> so we were able to more easily extend
> the condition to the negative comonotonicity,
> and study variations such as FEG-A and S-FEG.
>
> $\bullet$ **(Practical problem):**
> We do not have a concrete practical example that satisfies the comonotoncity,
>     but we believe that many practical problems
>     will likely satisfy such condition (at least locally).
>     To justify this claim,
>     we would like to further explain
>     the $\alpha\ge0$-interaction dominant condition in [10]
>     (yielding the comonotonicity).
>     A convex-concave function is $\alpha\ge0$-interaction dominant.
>     On the other hand, for any $\gamma$-weakly-convex-weakly-concave function,
>     the condition holds with $\alpha = -\gamma < 0$.
>     Its extreme case is $\phi(x,y) = -\frac{\gamma}{2}x^2 + \frac{\gamma}{2}y^2$,
>     where there is no interaction between $x$ and $y$.
>     Interestingly, a nonconvex-nonconcave function
>     becomes $\alpha\ge0$-interaction dominant,
>     when the second term in the condition is sufficiently positive definite.
>     In other words, the condition is satisfied
>     when the interaction term of the Hessian
>     $\nabla^2_{xy}\phi$ is dominating any negative curvature
>     in Hessians $\nabla_{xx}^2\phi$ and $-\nabla_{yy}^2\phi$.
>     In short, the $\alpha\ge0$-interaction dominant condition captures whether
>     there is a sufficient coupling between $x$ and $y$,
>     and we believe many practical (adversarial) applications
>     (at least locally or asymptotically) satisfy such general interaction condition.
>
> $\bullet$ **(S-FEG):**
> If you look it carefully, you will see that
> S-FEG is identical to FEG with $\rho=0$,
> except that it utilizes a noisy saddle gradient oracle.
> FYI, the condition $\rho\in\big(-\frac{1}{2L},\infty\big)$ in Algorithm 3
> should have been removed.
> We were not specifically aiming for
> S-FEG to have $O(1/k^2)$ rate of convergence (plus $\epsilon$ in Theorem 6.1),
> and this was a most straightforward stochastic analysis
> we were able to come up with.
> As mentioned in line 217-219, we are interested extending the stochastic result.
> We totally agree that studying the lower complexity bound
> in terms of the squared gradient norm
> for the minimax problems
> will be of significant interest.
> This, however,
> is beyond the scope of this paper,
> and we leave as future work.
>
> $\bullet$ **(Theorem 4.1):**
> Regarding the second last line of line 239,
> since $Fz_* = 0$, we get $\langle Fz_k , z_k-z_*\rangle \ge \rho||Fz_k||^2$ by the $\rho$-comonotonicity on $F$.
> We will let the readers know that its proof is given Section 7.1.
>
> $\bullet$ **(Typos):**
> We will fix typos, as suggested.

---

> > ### Comment · Reviewer_KwBn · 2021-08-29
> > **Keep my rating**
> >
> > Thanks for the answers. I think this is a solid work and am happy to keep the rating.
> >
> > Regarding that NeurIPS is a machine learning conference, however, I suggest the authors try to find some machine learning problems that satisfy their assumptions, to justify the significance of this work to the machine learning community. It would be good if the authors can provide some high-level insight behind the "algebraic trick."

---

> > > ### Author Response · Authors · 2021-08-31
> > > **Great to hear your positive and constructive comments**
> > >
> > > We agree that the current version might not attract a broader practical audience of NeurIPS. We were not able to come up with interesting practical applications under our setting at this point, like many other theory papers, but we will try our best to better justify the significance of this work to the ML audience. We will also spend more space to provide insights behind the "algebraic trick" in our revision.

---

### Decision · Program_Chairs · 2021-09-28

**Decision:**

Accept (Poster)

**Comment:**

The authors propose an extension of [Yoon and Ryu] along the function class of [Diakonikolas et al], which technically considers a bigger function class than [Mertikopoulos et al; extra mile]. The authors' provide a clean algorithm (FEG) and then try to make it practical via line-search. Finally, they provide a stylized example. On the theoretical side, the success of line-search is guaranteed provided that the parameters fall within an unknown threshold, which is a weakness. The authors are recommended to explain this weakness in their final version. Overall, it is a solid paper with some technical contributions over the existing optimization literature on weak MVIs.

**Consistency Experiment:**

NeurIPS has a long history of experimentation. In 2014, NeurIPS ran an experiment in which 10% of submissions were reviewed by two independent committees to quantify the randomness in the review process. This year, we repeated a variant of this experiment to see how the quality of the review process has changed over time.  This paper was part of the experiment and was therefore assigned to two committees (consisting of reviewers, an Area Chair, and a Senior Area Chair) that reached independent decisions.  If both committees made the same recommendation, this recommendation was followed. If a single committee recommended acceptance, the paper was accepted (with the exception of a few cases in which the other committee identified what we considered a fatal flaw, e.g., an error in a key result).

Both committees reached the same decision: **Accept (Poster)**

The other committee assigned to the paper recommended **Accept (Poster)**.  You can find the other set of reviews, along with any follow up discussion with the authors here:
https://openreview.net/forum?id=U7vVeHydyR